# NK cells promote cardiac cell death and regulate myelopoiesis in myocardial infarction

Raphael Cohen[1,19], Vincent Duval[1,19], Rida Al-Rifai[1], Sidrah Maryam[2], Icia Santos-Zas[3], Rayan Braik[1], Marc Diedisheim[4], Charlène Jouve[1], Simon Nicoletti[5], Sara Thietart[1], Théo Guyon[1], Luna Chetrit[1], Maria Kral[6,7], Yvonne Döring[6,7,8,9], Christian Weber[6,10,11], Alexandre Loupy[1], Orianne Domenge[12], Marine Livrozet[12], Jean-Sebastien Hulot[1], Ecem T. Sakalli[13], Giuseppe Rizzo[13], Antoine-Emmanuel Saliba[14,15], Marie Piollet[1,13], Patrick Bruneval[1], Alain Tedgui[1], Soraya Taleb[1], Olivier Hermine[5], Ian Wicks[16], Jean-Sébastien Silvestre[1], Sikander Hayat[2], Clement Cochain[1,13], Eric Vivier[17] & Hafid Ait-Oufella[1,18] ✉

Ischemic heart failure remains a major clinical challenge, underscoring the need to better understand post-infarction immune mechanisms and identify new therapeutic targets. Both innate and adaptive immunity contribute to adverse cardiac remodeling following myocardial infarction (MI), yet the role of cytotoxic cells such as natural killer (NK) cells remains poorly defined. Here, we show that after acute MI in mice, NK cells are recruited to the ischemic myocardium in a CCR2-dependent manner and become activated. Activated NK cells locally release granzyme B, promoting cardiomyocyte apoptosis, adverse ventricular remodeling, and impaired cardiac function. Genetic deletion or pharmacological depletion of NK cells reduces cardiomyocyte death, attenuates inflammation, limits myocardial injury, and improves cardiac function. In contrast, NK cell activation using an anti-NKG2A monoclonal antibody exacerbates ischemic heart failure. We further demonstrate that NK cells regulate bone marrow myelopoiesis through local GM-CSF production. Finally, we identify a distinct NK cellular and transcriptomic signature in human ischemic heart tissue at early stages. Together, these findings reveal a detrimental role for NK cells following acute MI and highlight NK cells as potential therapeutic targets to limit adverse cardiac remodeling.

Despite significant advances in the early management of acute myocardial infarction (MI), ischemia-related heart failure remains a persistent clinical and social burden, including a heightened risks of arrhythmias, recurrent hospitalizations, disabilities, and mortality[1]. Recent epidemiological evidence, including a study using nationwide medico-administrative data from the French National Hospitalization Database (2013–2018), underscores the rising incidence of ischemic heart failure, especially among young adults[2]. This trend underscores the need for intensified efforts to deepen our understanding of post-ischemic pathophysiology and develop treatments targeting specific pathways involved in cardiac tissue remodeling.

Several human and experimental studies have highlighted both beneficial and harmful effects of inflammation following acute MI and reperfusion injury[1]. Large numbers of immune cells are rapidly recruited to the heart to facilitate the clearance of dead cardiac myocytes and debris, and to orchestrate tissue repair[3]. However, blood-derived leukocytes also contribute to long-term cardiac damage. Recruited immune cells, activated by Danger/Damage Associated Molecular Patterns (DAMPS), release signaling molecules, including chemokines, cytokines, and reactive oxygen species, which contribute to detrimental remodeling of the heart by modulating the production and degradation of extracellular matrix proteins, and impairing the functions of local cardiac and non-cardiac cells[4]. Notably, recent research from our group has identified CD8[+] T cells, a subset of T lymphocytes with cytotoxic activity, as key players in post-ischemic cardiac remodeling, facilitating cardiomyocyte death through the local release of Granzyme B[5,6].

Natural Killer (NK) cells represent another powerful cytotoxic population. NK cells represent the largest subset of the innate lymphoid cell (ILC) family and share many common features with ILC1 cells[7,8]. Despite lacking somatic rearrangements of antigen receptors, NK cells are responsive to multiple immune signals necessary for pathogen defense and tissue homeostasis[9]. However, the role of NK cells in post-ischemic cardiac remodeling remains poorly understood.

Here, utilizing a combination of gain-of-function and loss-of-function approaches in a murine model of MI, we demonstrated that NK cells are activated and are recruited to the ischemic heart in a CCR2-dependent manner. Locally, NK cells induce cardiomyocyte death through the release of Granzyme B, which exacerbates myocardial inflammation, tissue injury, and reduces myocardial function. Additionally, we revealed that NK cells modulate myelopoiesis in the bone marrow (BM) through the release of granulocyte-macrophage colony-stimulating factor (GM-CSF). Furthermore, immunotherapy based on the administration of neutralizing antibodies targeting NK cells promoted cardiac repair in a murine model of MI.

## Results

### NK cells are recruited and activated in the ischemic heart after acute MI

First, we investigated the trafficking of NK cells following acute MI. To achieve this, we employed an experimental model of MI in C57BL/6 J mice, induced by permanent coronary artery ligation. Subsequently, we analyzed cell suspensions obtained from digested hearts at various time points following the onset of ischemia using flow cytometry. Our observations revealed accumulation of CD3[-]NK1.1[+]NKp46[+] NK cells in the injured myocardium, starting as early as day 3 and peaking at day 7 after MI (Fig. 1a, b & Supplementary Fig. 1). Notably, the infiltration of NK cells was around twice as high as that of CD8[+] T cells (Supplementary Fig. 2a). Further immunofluorescent staining and transcriptomic studies (Fig. 1c, d & Supplementary Fig. 3) confirmed the heightened presence of NK cells in both peri-infarct and infarcted areas of the heart following MI, as compared to sham-operated animals. To further characterize NK cell populations in the context of MI, we reanalyzed previously published data on cellular indexing of transcriptomes and epitopes by sequencing (CITE-seq)[10] of total CD45[+] cells from control and ischemic hearts at day 5 post-MI[11]. Cells corresponding to T and NK cells were extracted and separately reclustered. The infiltrating NK cells predominantly exhibited a mature (Itgam[+]) CD11b[+] phenotype (Fig. 1e, f), along with a cytotoxic transcriptomic signature, as evidenced by enrichment for Gzma, Gzmb, and Perforin mRNA within this population (Fig. 1g, h). Mature NK cells also express high Itga4 mRNA (CD49d), an integrin involved in adhesion (Supplementary Fig. 4). NK cells within ischemic hearts were found to express Ncr1 (encoding NKp46), Klrc1 (encoding NKG2A), and Klrk1 (encoding NKG2D) receptors, which are crucial for regulating their activation

(Fig. 1i–l). We then investigated the underlying mechanisms governing NK cell trafficking, focusing on CCR2 and CCR5, as both chemokine receptors are expressed by infiltrating NK cells (Fig. 1m, n & Supplementary Fig. 5ab). Deletion of Ccr2, while not impacting CCR5 expression, resulted in impaired NK cell recruitment to the heart in the context of MI (Supplementary Fig. 5c–e). Conversely, deletion of Ccr5, which is associated with an up-regulation of CCR2, led to an increase in NK cell infiltration within the ischemic heart tissue (Supplementary Fig. 5f–i). Given that global Ccr2 deletion affects multiple immune cell populations, we performed additional experiments to specifically validate the critical role of CCR2 for NK cell recruitment to ischemic heart tissue. To this end, Nkp46[iCre+/−]R26R[DTA] (referred to as NK[KO]) mice were repopulated with purified Wild-type or Ccr2[−/−] NK cells 3 days before MI (Fig. 1o). The purity of NK cells and the NK cell deficiency in NK[KO] mice were confirmed by flow cytometry (Supplementary Fig. 5j-6). As depicted in Fig. 1p, at day 3 post-MI, NK cell frequencies were comparable in the blood across groups but were significant lower in the hearts of MI-operated mice repopulated with Ccr2[−/−] NK cells (Fig. 1q, r). Collectively, these findings suggest that mature cytotoxic NK cells are recruited into the myocardium following MI partly via CCR2, thus implicating a potential role for NK cell-mediated immune responses in this setting.

### NK depletion or deficiency prevents deleterious adverse ventricular remodeling after acute MI in mice

To directly investigate the impact of NK cells on cardiac remodeling following MI, we administered an NK1.1 monoclonal antibody[12] to deplete NK cells one hour after the onset of MI. This treatment resulted in a rapid and substantial depletion of NK cells ( > 95%) within 6 h post-injection in the peripheral blood, bone marrow (BM), spleen and cardiac tissue (Supplementary Fig. 7a–c & Fig. 2a). NK1.1 mAb had no impact on CD8[+] T cells (Supplementary Fig. 7d). It is noteworthy that NK1.1 mAb treatment also effectively depleted NKT cells, despite their being a small population in the ischemic heart (Supplementary Fig. 2b). Complete NK cell depletion was confirmed by immunofluorescence in the ischemic heart (Fig. 2b). Echocardiography performed 21 days post-MI revealed a significant improvement in left ventricular (LV) ejection fraction (Fig. 2c) and cardiac output (Fig. 2d) in mice subjected to NK1.1 mAb-induced NK cell depletion, compared to isotype-treated control mice. The improvement in cardiac function following NK1.1 mAb treatment was accompanied by a reduction in adverse LV remodeling. Both infarct size (Fig. 2e) and interstitial fibrosis (Fig. 2f), as assessed by collagen content, were significantly diminished in NK1.1 mAb-treated mice ($p = 0.004$ and $p = 0.036$, respectively) compared to isotype-treated control animals at day 21 post-MI. Importantly, the protective effect of NK cell depletion was evident in female as well as male C57BL/6 J mice (Supplementary Fig. 8). NK cell depletion correlated with decreased collagen synthesis, as evidenced by reductions in Col1a1 and Col3a1 mRNA levels (Fig. 2g).

To further corroborate the pathogenic role of NK cells in post-ischemic cardiac remodeling, we employed Nkp46[iCre+/−]R26R[DTA] (referred to as NK[KO]) mice, littermate Nkp46[iCre-] R26R[DTA] mice served as controls (NK[WT]). At day 21 post-MI, NK[KO] mice had improved LV systolic function (Fig. 2i) and higher cardiac output (Fig. 2j), compared to control NK[WT] mice. Additionally, both infarct size (Fig. 2k) and interstitial fibrosis (Fig. 2l) were significantly decreased in NK-deficient mice, further emphasizing the pathogenic role of NK cells in adverse post-ischemic cardiac remodeling.

### NK activation aggravates post-ischemic cardiac damage in experimental MI

To further evaluate the role of NK cells in post-ischemic cardiac remodeling, we utilized a complementary in vivo gain-of-function approach involving a neutralizing anti-NKG2A monoclonal antibody (or IgG isotype control) in C57BL/6 J mice (Fig. 3a)[12]. NKG2A is an

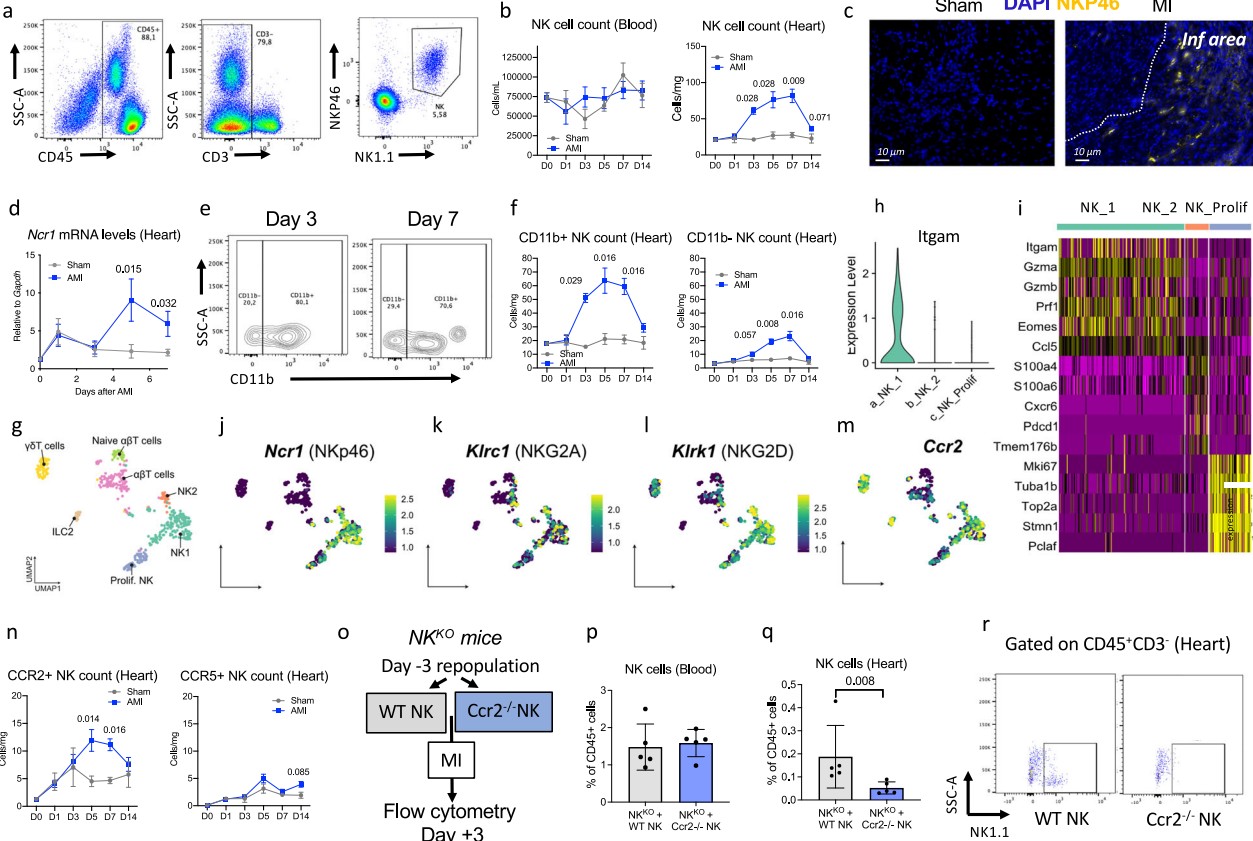

**Fig. 1 | NK are recruited to the ischemic tissue after myocardial infarction.**
**a** flow cytometry characterization of CD45+CD3-NK1.1+NKP46+ NK cells in the blood at day 3 post-MI. **b** Kinetic of NK cells in the blood and in the heart, at baseline and post-MI (Blue, $n = 8/5/5/5/5$ respectively at day 0/1/3/5/7/14) and Sham operated mice (Gray, $n = 8/3/3/3/3$ respectively at day 0/1/3/5/7/14). **c** Representative examples of NKP46+ NK cell staining in the heart of C57BL/6 J mice following coronary ligation (MI) or sham surgery at day 3. Scale bar 10 μm. **d** Kinetic of *Ncr1* mRNA levels in the ischemic heart tissue quantified by qPCR after MI (Blue) or Sham surgery (Gray) ($n = 8/5/5/5/5$ respectively at day 0/1/3/5/7). **e** Flow cytometry characterization of NK cell according to CD11b expression at day 3 and day 7 post-MI. **f** Kinetic of CD11b+ and CD11b- NK cell subsets in the heart, at baseline and post-MI (Blue, $n = 8/5/5/5/5$ respectively at day 0/1/3/5/7/14) and Sham operated mice (Gray, $n = 8/3/3/3/3$ respectively at day 0/1/3/5/7/14). **g** UMAP visualization of scRNA-seq profiles of non-myeloid immune cells extracted from aortas. **h** gene expression levels of *Itgam* in NK subsets identified by Sc-RNA seq at day 5 after MI.

**i** Transcriptomic profile of 3 subsets of NK cells clustered in ischemic heart tissue at day 5 post-MI. **j** expression of *Ncr1* transcript in the populations. **k** Expression of *Klrc1* transcript in the populations. **l** Expression of *Klrk1* transcript in the populations. m, expression of *Ccr2* transcript in the populations. n, kinetic of CCR2+ and CCR5+ NK cells in the heart at baseline and after MI (Blue, $n = 8/5/5/5/5$ respectively at day 0/1/3/5/7/14) and Sham operated mice (Gray, $n = 8/3/3/3/3$ respectively at day 0/1/3/5/7/14). **o** Experimental protocol NK cell repopulation.
**p** CD45+CD3-NK1.1+ NK cell quantification in the blood at day 3 after MI, repopulation being done with WT ($n = 5$) or Ccr2−/−($n = 5$) purified NK cells.
**q** CD45+CD3-NK1.1+ NK cell quantification in the heart at day 3 after MI, repopulation being done with WT ($n = 5$) or Ccr2−/−($n = 5$) purified NK cells. **r** Representative flow cytometry pictures of WT and *Ccr2−/−* NK cell repopulation of $NK^{KO}$ mice. *P* values were calculated using two-tailed Mann–Whitney test. Inf, Infarct; MI, myocardial infarction. Data are presented as mean values ± SEM.

inhibitory receptor, and its blockade enhances NK cell activity. Increased activation of NK cells following NKG2A mAb treatment was confirmed by flow cytometry in the spleen and BM at day 3 after acute MI (Supplementary Fig. 9). NKG2A mAb treatment increased total CD45+ leukocytes, but had no impact on NK cell numbers in the heart at day 5 after MI (Fig. 3b–d). However, NKG2A mAb treatment significantly aggravated cardiac damage induced by acute ischemia, with increased heart weight (Fig. 3e), greater LV dilation (Fig. 3f), and reduced LV ejection fraction (Fig. 3g) at day 21 compared to isotype-treated control mice. The worsening of cardiac dysfunction following NKG2A mAb treatment was associated with exacerbated adverse LV remodeling. Both infarct size (Fig. 3h) and interstitial fibrosis (Fig. 3i & Supplementary Fig. 10), as assessed by collagen content, were significantly increased in NKG2A mAb-treated mice compared to isotype-treated control animals at day 21 post-MI. Consistent with these findings, cardiac *Col1a1* mRNA levels in heart tissue were significantly increased in the NKG2A group (Fig. 3j).

Given the NKG2A receptor is also expressed by CD8+ T cells, the pathogenic effects of the NKG2A mAb may not be solely due to NK cell activation. To specifically evaluate the impact of NK cell activation in the context of MI, we repeated anti-NKG2A monoclonal antibody or IgG isotype treatments in CD8-depleted C57BL/6 J mice (Fig. 3k). CD8 depletion was confirmed by flow cytometry (Fig. 3l). As shown in Fig. 3m–o, in the absence of CD8+ T cells, NKG2A mAb still aggravated post-ischemic cardiac damage, resulting in reduced LV systolic function, larger infarct size and increased fibrosis, compared to isotype treatment.

### Pathogenic activity of NK cells after MI is not due to NKp46 receptor

Our findings indicated that NK cells play a role in harmful post-ischemic cardiac remodeling. We then examined which receptors govern NK cell pathogenic activity, initially focusing on NKp46, as its role in cardiac remodeling is not well understood. Previous studies

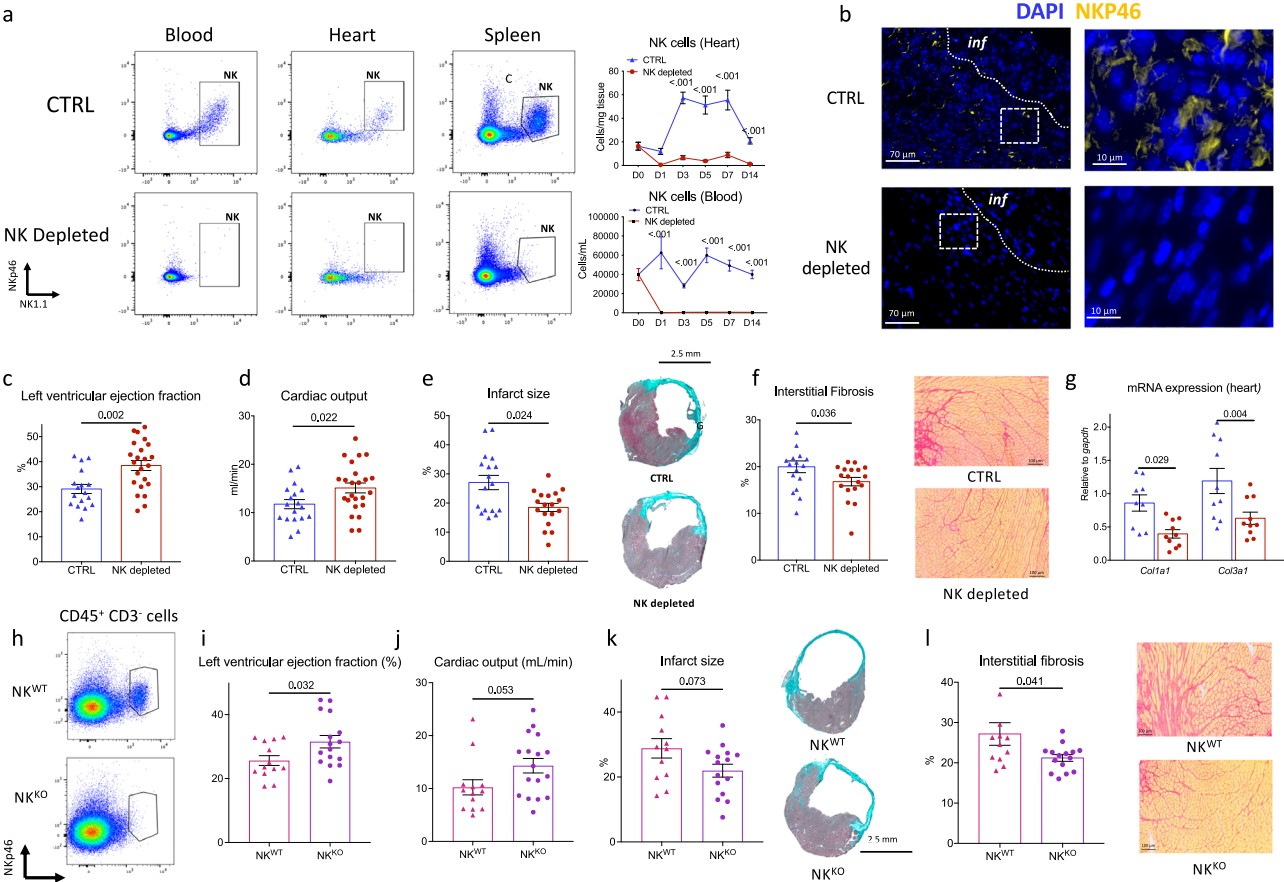

**Fig. 2 | NK cell depletion or deficiency improves heart function and reduces infarct size. a** Representative examples and quantitative analysis of NK cell staining in the blood, the heart, and the spleen of C57BL/6 J mice treated with isotype control (CTRL, blue, *n* = 7/5/6/8/7 respectively at day 0/1/3/5/7/14) or with the NK1.1 mAb (NK depleted, red, *n* = 7/6/7/6/8/7 respectively at day 0/1/3/5/7/14). **b** Representative examples (left) of NK cell staining (NKP46, Yellow) in the heart of C57BL/6 J mice depleted or not of NK cells at day 3 (Representative of 6 samples) scale bar 70 µm. Echocardiography analysis after anti-NK1.1 therapy. We measured left ventricle (LV) ejection fraction (**c**) (CTRL blue, *n* = 17 and NK Depleted red *n* = 25) and cardiac output (**d**) at day 21 (CTRL blue, *n* = 18 and NK Depleted red *n* = 23). **e** Representative photomicrographs and quantitative analysis of infarct size evaluation evaluated by Masson trichrome staining, in the 2 groups of mice at day 21 (CTRL *n* = 17 and NK depleted *n* = 18) Scale bar 2.5 mm. **f** Representative photomicrographs and quantitative analysis of myocardial fibrosis evaluated by Sirius Red staining, in the 2 groups of mice at day 21 (CTRL *n* = 17 and NK depleted *n* = 18). Scale bar 100 µm. **g** Representative histograms of mRNA levels of *Col1a1* and *Col3a1* within the injured myocardium on day 5 post-MI (*n* = 10/group; CTRL blue and NK depleted Red). h, validation of NK cell deficiency in NK$^{KO}$ mice using flow cytometry in the spleen. Left ventricle (LV) ejection fraction (**i**) and cardiac output (**j**) measured by echocardiography (NK$^{WT}$ pink, *n* = 13 and NK$^{KO}$ purple *n* = 16). **k** Representative photomicrographs and quantitative analysis of infarct size evaluation evaluated by Masson trichrome staining, in the 2 groups of mice (NK$^{WT}$ *n* = 13 and NK$^{KO}$ *n* = 15). Scale bar 2.5 mm. **l** Representative photomicrographs and quantitative analysis of myocardial fibrosis evaluated by Sirius Red staining, in the 2 groups of mice (NK$^{WT}$ *n* = 12 and NK$^{KO}$ *n* = 15) Scale bar 100 µm. **a**, **g**: 2 pooled experiments; **c**–**f** and **i**–**l**: 3 pooled experiments. *P* values were calculated using two-tailed Mann–Whitney test. Inf, Infarct. Data are presented as mean values ± SEM.

have shown that NKp46 regulates the growth of tumors such as melanoma[13], lymphoma[14], or carcinoma. To study NKp46 in vivo, we utilized *Ncr1* knockout mice, in which the *Ncr1* gene (coding for NKp46) was replaced by a green fluorescent protein (Gfp) reporter (*Ncr1*$^{gfp/gfp}$)[15] (Supplementary Fig. 11a). NK cell counts in the circulation were similar between control *Ncr1*$^{+/+}$ and *Ncr1*$^{gfp/gfp}$ mice (Supplementary Fig. 11b). Notably, *Ncr1* deficiency did not affect NK cell infiltration into the ischemic heart at day 5 post-MI (Supplementary Fig. 11c) and had no significant impact on cardiac function (Supplementary Fig. 11d), infarct size (Supplementary Fig. 11e) or interstitial fibrosis (Supplementary Fig. 11f) at day 21. Taken together, these results suggest that the pathogenic activity of NK cells in the context of MI do not require the engagement of the NKp46 receptor.

**NK cells aggravate cardiac cell death through Granzyme B**

We next explored the mechanisms through which NK cells influence cardiac remodeling and function, focusing on their well-known cytotoxic activity, which has been documented in cancers and viral infections[16]. ScRNA-seq analysis showed that infiltrating NK cells express Granzyme A, Granzyme B, and Perforin (Fig. 4a). Following MI, *Gzma* mRNA levels in the heart remained relatively constant, but *Gzmb* levels significantly increased between days 3 and 7, coinciding with the infiltration of CD11b$^+$ NK cells (Fig. 4b, c). Depletion of NK cells with the NK1.1 mAb treatment resulted in reduced *Gzmb* mRNA levels in ischemic heart tissue (Fig. 4d) and a complete absence of Gzmb$^+$ NKp46$^+$ NK cells (Fig. 4e). Of note, a population of GzmB$^+$ NKp46$^-$ cells, corresponding to CD8$^+$ T cells, remained detectable in the ischemic heart (Fig. 4e). The decrease in cardiac Granzyme B content was associated with a significant reduction in both TUNEL+ cells and infarct size 3 days post-MI (Fig. 4f & Supplementary Fig. 12). In contrast, NK cell activation using anti-NKG2A treatment increased *Gzmb* mRNA levels in ischemic heart tissue (Fig. 4g), elevated GzmB$^+$ NKp46$^+$ NK cell number (Fig. 4h), and led to a substantial increase in both TUNEL+ cells and infarct size 3 days post-MI (Fig. 4i & Supplementary Fig. 12). The cytotoxic role of NK cells was confirmed in NK-deficient mice, where tissue necrosis was significantly reduced (Fig. 4j).

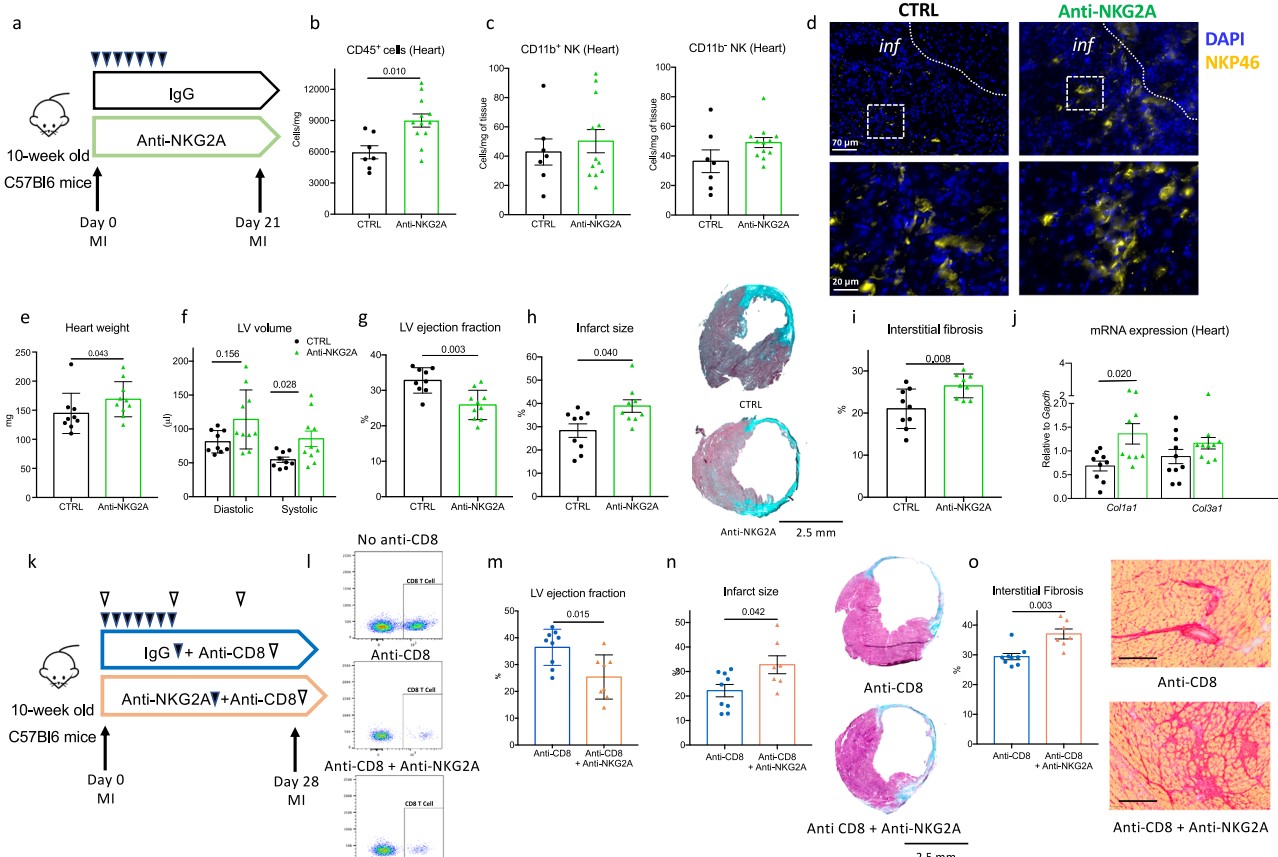

**Fig. 3 | NK cell activation aggravates heart dysfunction and increases infarct size. a** Experimental protocol with isotype (Black) or anti-NKG2A mAb (Green) treatment. **b** CD45⁺ leukocyte count in the ischemic heart at day 5 after MI ($n = 7$ CTRL and $n = 12$ Anti-NKG2A). **c** NK subsets count in the ischemic heart at day 5 post-MI ($n = 7$ CTRL and $n = 12$ Anti-NKG2A). **d** Representative examples of NKp46⁺ NK cell staining (Yellow) in the heart at day 5 after MI (Representative of $n = 6$ mice per group), scale bar 70 μm upper and 20 μm lower. **e** Heart weight at day 21 after MI ($n = 9$ CTRL and $n = 10$ Anti-NKG2A). **f** Telediastolic and telesystolic LV volume measured by echocardiography at day 21 post-MI ($n = 9$ CTRL and $n = 10$ Anti-NKG2A). **g** LV ejection fraction at day 21 ($n = 9$ CTRL and $n = 10$ Anti-NKG2A). **h** Representative photomicrographs and quantitative analysis of infarct size evaluation evaluated by Masson trichrome staining at day 21 ($n = 9$/group), scale bar 2.5 mm. **i** representative photomicrographs and quantitative analysis of myocardial fibrosis evaluated by Sirius Red staining, in the 2 groups of mice at day 28 ($n = 9$/

group), scale bar 100 μm. **j** Representative histograms of mRNA levels of *Col1a1* and *Col3a1* within the injured myocardium on day 5 post-MI ($n = 9$/group; CTRL black and anti-NKG2A Green). **k** experimental protocol with isotype (Blue) or anti-NKG2A mAb (Orange) treatment in CD8 depleted mice. **l** Validation of CD8 depletion by blood cytometry in the blood at day 3 (representative of 8 mice/group). **m** LV ejection fraction at day 21 ($n = 9$ CD8 depleted and $n = 8$ CD8 depleted + Anti-NKG2A). **n** Representative photomicrographs and quantitative analysis of infarct size evaluation evaluated by Masson trichrome staining, in the 2 groups of mice at day 28 ($n = 9$ CD8 depleted and $n = 7$ CD8 depleted + Anti-NKG2A); Scale bar 2.5 mm. **o** Representative photomicrographs and quantitative analysis of myocardial fibrosis evaluated by Sirius Red staining, in the 2 groups of mice at day 21 ($n = 9$ CD8 depleted and $n = 7$ CD8 depleted + Anti-NKG2A); Scale bar 100 μm. *P* values were calculated using two-tailed Mann–Whitney test. Inf, Infarct. Data are presented as mean values ± SEM.

To further investigate the role of NK cell-derived Granzyme B, we reconstituted NK-deficient mice with purified NK cells from wild-type or *Gzmb⁻/⁻* mice (Fig. 4k) 48 h before MI. Immunofluorescent staining confirmed the successful transfer of both wild-type or *Gzmb⁻/⁻* NK cells (Supplementary Fig. 13). Transferring wild-type NK cells into NK-deficient mice resulted in increased heart tissue necrosis 3 days post-MI, whereas the transfer of *Gzmb⁻/⁻* NK cells did not result in increased necrosis (Fig. 4l). These findings suggest that Granzyme B has direct cytotoxic effects on cardiomyocytes. To test this hypothesis, we co-cultured mouse cardiomyocytes with purified wild-type or *Gzmb⁻/⁻* NK cells (Fig. 4m–o & Supplementary Fig. 14). After 24 h, NK cells were removed, and cardiomyocyte apoptosis was monitored for an additional 24 h using a caspase-3 fluorescent dye. Incubation with non-activated NK cells only modestly affected cardiomyocyte survival, while incubation with activated NK cells caused cardiomyocyte apoptosis, with the effect being more pronounced at higher NK/cardiomyocyte ratios (Fig. 4o). The cytotoxicity of NK cells was significantly reduced when Granzyme B was deficient, as shown by decreased

apoptosis rate in cardiomyocytes (Fig. 4p). Finally, to assess the human in vitro relevance of our findings, we co-cultured human 3D cardiac rings with purified human NK cells, and measured contractility parameters before and 12 h after co-culture (Fig. 4q & Supplementary Fig. 15a). Non-activated NK cells did not affect 3D cardiac ring function. However, activated NK cells which aggregated on rings, impaired both organoid contractility and relaxation in a dose-dependent manner (Fig. 4r & Supplementary Fig. 15b). Overall, these results strongly suggest that infiltrating NK cells impair cardiomyocyte function and survival in the context of MI through the release of Granzyme B.

## NK cells orchestrate myeloid cell trafficking and inflammatory responses

Modulating NK cells had a significant impact on the inflammatory profile of the heart at day 7 post-MI. In NK-depleted mice, the *mRNA* levels of *Il-1b and Tnf* were significantly lower in infarct hearts compared to controls (Fig. 5a & Supplementary Fig. 16a), along with a marked reduction in metalloproteinase activity (Fig. 5b-c). NK cell

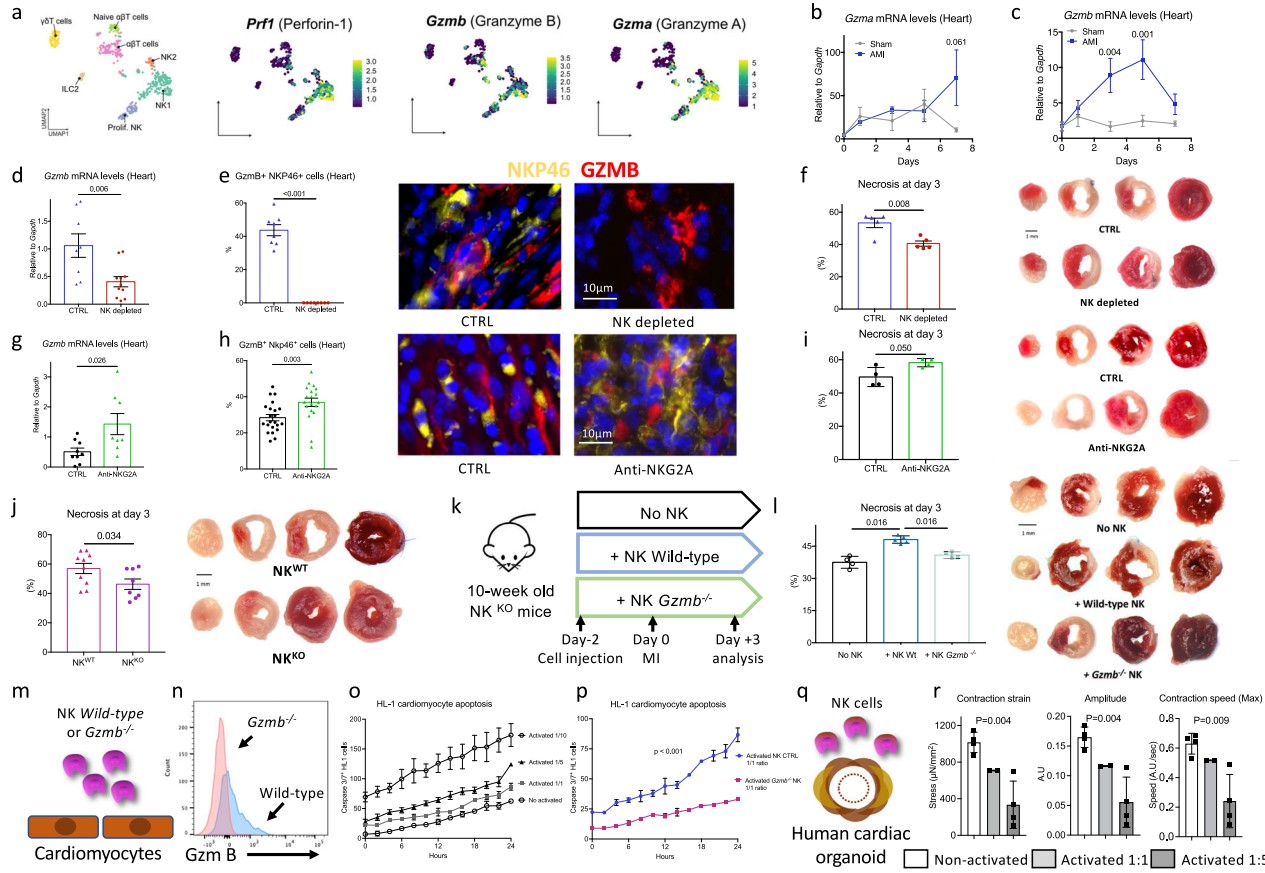

**Fig. 4 | NK cells promote tissue necrosis within the ischemic heart through the release of Granzyme B. a** UMAP visualization of scRNA-seq profiles of non-myeloid immune cells extracted from aortas; expression of *Perforin, Granzyme b and Granzyme a* transcript in the populations. **b, c** Kinetic of *Granzyme a and Granzyme b* mRNA in the ischemic heart tissue and Sham operated mice at days 0, 1, 3, and 7 post-MI (*Gzma* n = 4/ timepoint and *Gzmb* n = 5/time point). d, representative histograms of mRNA levels of *Granzyme b* within the injured myocardium on day 5 post-MI in CTRL (Blue, n = 8) and NK (Red, n = 11) depleted mice. **e** Quantitative analysis and representative examples (right) of Granzyme B + NKp46+ NK cell content in the ischemic heart of C57BL/6 J mice with or without NK depletion (n = 8/ group); Scale bars 10 μm. **f** Quantitative analysis and representative photomicrographs of heart tissue necrosis (white) after Triphenyltetrazolium Chloride staining, in the 2 groups of mice at day 3 post-MI (n = 5/group). **g** Representative histograms of mRNA levels of *Granzyme b* within the injured myocardium on day 5 after MI in isotype (Black, n = 9) and Anti-NKG2A (Green, n = 8) treated mice. **h** Quantitative analysis and representative examples (right) of Granzyme B⁺ NKp46⁺ NK cell content in the ischemic heart of C57BL/6 J mice treated with isotype (Black, n = 22) and Anti-NKG2A mAb (Green, n = 19); Scale bars 10 μm. **i** Quantitative analysis and representative photomicrographs of heart tissue necrosis (white) after Triphenyltetrazolium Chloride staining, in the 2 groups of mice at day 3 after MI (n = 4/group). **j** Quantitative analysis and representative photomicrographs of heart

tissue necrosis (white) after Triphenyltetrazolium Chloride staining, in NK^WT (pink, n = 10) and NK^KO (Purple, n = 8) mice at day 3 after MI. **k** Experimental protocol of NK repopulation in NK deficient mice. **l** Quantitative analysis and representative photomicrographs of heart tissue necrosis (white) after Triphenyltetrazolium Chloride staining, in the 3 groups of mice at day 3 after MI (No NK supplementation n = 4, wilt-type NK cells n = 5, *Granzyme b* deficient NK cells n = 4). **m** Purified non-activated or activated NK cells were co-cultured with cardiomyocytes at different ratio (Cardiomyocyte/NK cells) during 24 h before their removal. Apoptotic cardiomyocytes labeled with an active caspase-3 fluorescent dye was monitored during 24 h. **n** Granzyme B MFI in purified NK cells using flow cytometry. **o** Quantification of cardiomyocyte apoptosis overtime with different Cardiomyocyte/wild-type NK cells ratios (n = 4/condition/timepoint). **p** Quantification of cardiomyocyte apoptosis overtime after co-culture with wild-type (Blue) or Granzyme b deficient (pink) NK cells, Cardiomyocyte/wild-type NK cell ratio at 1 (n = 4/condition/ timepoint). **q** Co-culture of purified human NK cells with human 3D cardiac rings. **r** Contractility parameters of human 3D cardiac rings after 12 h of co-culture with non-activated NK cells 1:1 ratio (n = 4 rings), activated NK cells 1:1 ratio (n = 2 rings), activated NK cells 1:5 ratio (n = 4 rings), Data are presented as mean values ± SEM, P values were calculated using two-tailed Mann–Whitney test, Kruskal-Wallis test (**l**) or ANOVA test (**r**).

depletion also limited monocyte recruitment in the heart at days 5 and 7 post-MI (Fig. 5d and Supplementary Figs. 17, 18) and decreased the content of CCR2⁺ macrophages at day 7 (Supplementary Fig. 18). To further evaluate the impact of NK depletion on the transcriptomic phenotype of myeloid cells, scRNA-seq was performed on heart CD45⁺ cells at day 5 post-MI in animals treated or not with anti-NK1.1 depleting antibody (Supplementary Fig. 19a). Following clustering and extraction of myeloid cells populations, we identified monocytes, 5 macrophages subsets, resident macrophages, and 3 neutrophil subsets (Supplementary Fig. 19b-c). Effective NK depletion in anti NK1.1-treated mice was confirmed (Supplementary Fig. 19d). Using pseudobulk

analysis, we identified differentially expressed genes in monocyte and macrophage populations after NK cell depletion and assessed their expression across total cells per group. Of note, *Cd36* and *Hdac9* were upregulated, whereas *P2rx7*, a gene involved in inflammasome activation, was downregulated in the NK depleted group (Supplementary Fig. 19e). Moreover, *Tnf* expression was reduced in the main populations of monocytes, macrophages and neutrophils in the depleted group (Supplementary Fig. 19f). Altogether, these results indicate that NK cell depletion promotes a shift in macrophages toward a less inflammatory phenotype. In contrast, anti-NKG2A treatment, which activates NK cells, drives a shift toward a more inflammatory cardiac

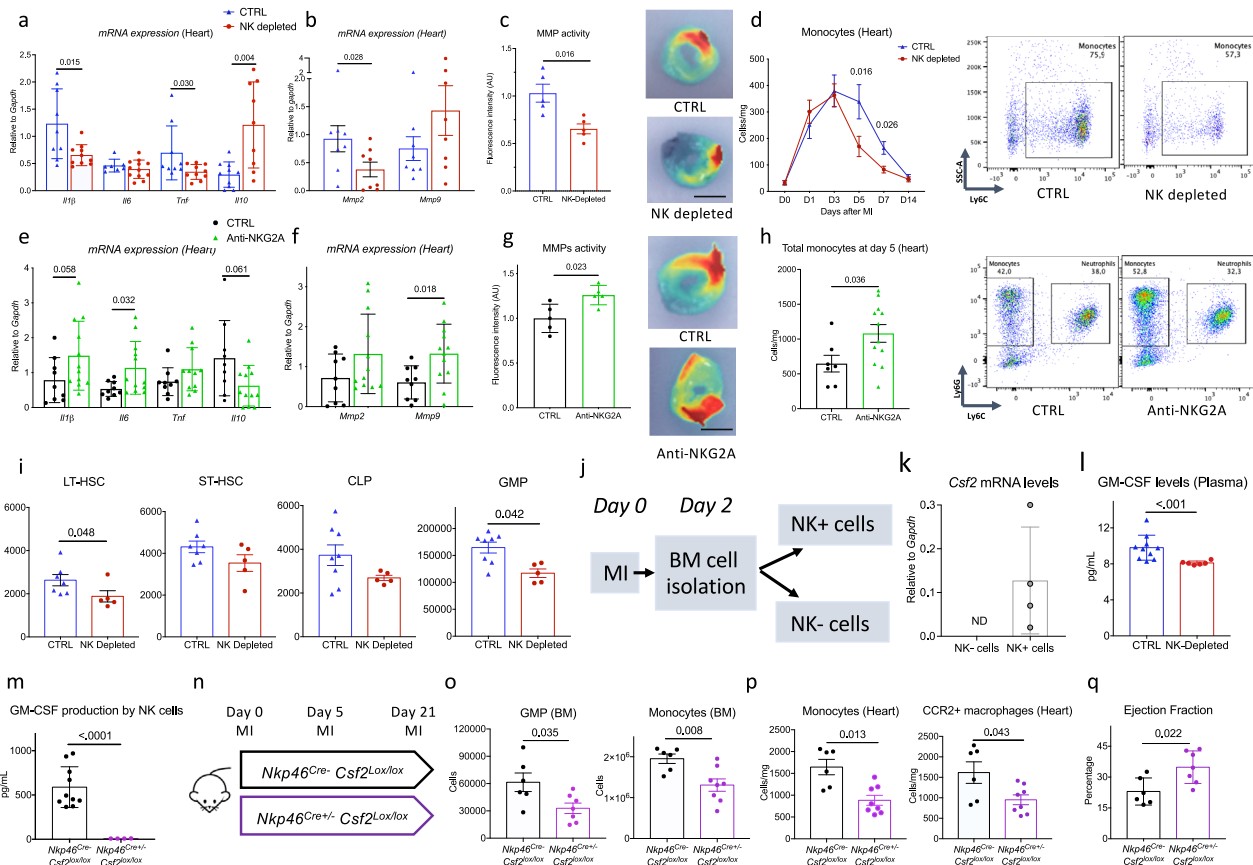

**Fig. 5 | NK cells modulate local innate inflammatory responses through GM-CSF production. a** Representative histograms of mRNA levels of *Il-1β, Il-6, Tnf,* and *Il-10* within the ischemic myocardium on day 7 after MI in mice with ($n$ = 10) or without ($n$ = 9) NK depletion. **b** Representative histograms of mRNA levels of *Mmp2 and Mmp9* within the ischemic myocardium on day 7 after MI in mice with or without NK depletion ($n$ = 8/group). **c** Quantification (left) and representative photomicrographs (right) of matrix metalloproteinase (MMP)-sense 680 activity in the ischemic heart measured by ex vivo reflectance epifluorescence imaging at day 7 ($n$ = 5/group), Scale bar 2.5 mm. **d** Quantification (left) and representative flow pictures (right) of monocytes in the heart, following MI in controls (blue, $n$ = 7/6/6/6/7 respectively at day 0/1/3/5/7/14) or NK depleted mice (red, $n$ = 7/6/7/5/6/7 respectively at day 0/1/3/5/7/14). **e**, representative histograms of mRNA levels of *Il-1β, Il-6, Tnf* and *Il-10* within the ischemic myocardium on day 7 after MI in mice treated with isotype ($n$ = 9) or anti-NKG2A mAb ($n$ = 12). **f** Representative histograms of mRNA levels of *Mmp2 and Mmp9* within the ischemic myocardium on day 7 after MI in mice treated with isotype ($n$ = 9) or anti-NKG2A mAb ($n$ = 12). **g** Quantification (left) and representative photomicrographs (right) of matrix metalloproteinase (MMP)-sense 680 activity in the ischemic heart measured by ex vivo reflectance epifluorescence imaging at day 7 ($n$ = 5/group), Scale bar

2.5 mm. **h** Quantification (left) and representative flow pictures (right) of monocytes in the heart at day 5 MI in isotype ($n$ = 7) or anti-NKG2A-treated mice (red, $n$ = 12). **i** Number of long-term (LT) and short-term (ST) hematopoietic stem cells (HSC), Common Lymphoid progenitors (CLP) and Granulocyte/macrophage progenitors (GMP), in the bone marrow (BM) (CTRL $n$ = 7, NK Depleted $n$ = 5) at Day 3 after MI. j, experimental protocol. **k** *Csf2* mRNA levels in NK+ and NK- cells in the BM at day 2 after MI ($N$ = 4/group). **l** GM-CSF levels in the plasma of CTRL and NK depleted mice at day 3 after MI (CTRL $n$ = 10; NK depleted, $n$ = 6). **m** GM-CSF levels in the supernatant of stimulated purified NK cells from *Nkp46^{Cre-}Csf2^{lox/lox}* and *Nkp46^{Cre+/-} Csf2^{lox/lox}* mice (ELISA, $n$ = 10 Cre- $n$ = 5 Cre +). n, experimental protocol, with analysis of leukocyte populations at 5 after MI and analysis of ejection fraction at day 21. o, quantification of GMP and monocytes in the BM of control *Nkp46^{Cre-}Csf2^{lox/lox}* and *Nkp46^{Cre+/-} Csf2^{lox/lox}* mice at day 5 after MI (Cre- $n$ = 6, Cre+ $n$ = 7). **p** Quantification of monocytes and CCR2+ macrophages in the heart of control *Nkp46^{Cre-} Csf2^{lox/lox}* and *Nkp46^{Cre+/-} Csf2^{lox/lox}* mice at day 5 after MI (Cre- $n$ = 6, Cre+ $n$ = 8). **q** Ejection fraction at day 21 after MI in control *Nkp46^{Cre-} Csf2^{lox/lox}* ($n$ = 6) and *Nkp46^{Cre+/-} Csf2^{lox/lox}* mice ($n$ = 7). Data are presented as mean values ± SEM, *P* values were calculated using two-tailed Mann–Whitney test.

phenotype, characterized by higher *Il-1b, Il-6 mRNA* levels, lower *Il-10 mRNA* levels (Fig. 5e), and greater local metalloproteinase activity (Fig. 5f-g). Consistently, Anti-NKG2A-mediated NK activation increased the recruitment of neutrophils and monocytes (Fig. 5h & Supplementary Fig. 18f-g) in the heart at day 5 post-MI and increased the number of CCR2+ macrophages at day 7 (Supplementary Fig. 18h).

We next examined how NK cell modulation affected monocyte content in the heart. We found that NK cell depletion significantly reduced monocyte counts in the spleen and BM within the first days post-MI (Supplementary Fig. 18d-e). In addition, granulocyte/macrophage progenitor cell (GMP) populations in the BM were reduced in NK-depleted mice (Fig. 5i & Supplementary Fig. 17). NK modulation did not affect *Ccl2, Ccl3, Cxcl1* and *CxcL2* gene expression in the heart (Supplementary Fig. 16). Based on these findings and previous studies

showing that NK cells produce GM-CSF[17], we hypothesized that NK cells control BM myelopoiesis through GM-CSF production during MI. We confirmed that NK cells are an important source of GM-CSF in the BM of MI-operated mice (Fig. 5j–k) and observed decreased plasma levels of GM-CSF at day 3 post-MI in NK-depleted mice compared to isotype-treated controls (Fig. 5l).

To specifically investigate the role of GM-CSF produced by NK cells in myelopoiesis during cardiac ischemia, we crossbred mice carrying an *Nkp46 Cre* allele with mice carrying a floxed *Csf2* allele. GM-CSF deficiency was confirmed by ELISA in the supernatant of stimulated purified NK cells from *Nkp46^{Cre+/-}Csf2^{lox/lox}* mice (Fig. 5m). Monocytes and neutrophils populations were similar between *Nkp46^{Cre+/-}Csf2^{lox/lox}* and littermate *Nkp46^{Cre-}Csf2^{lox/lox}* control mice at baseline as well as cardiac function (Supplementary Fig. 20). MI was

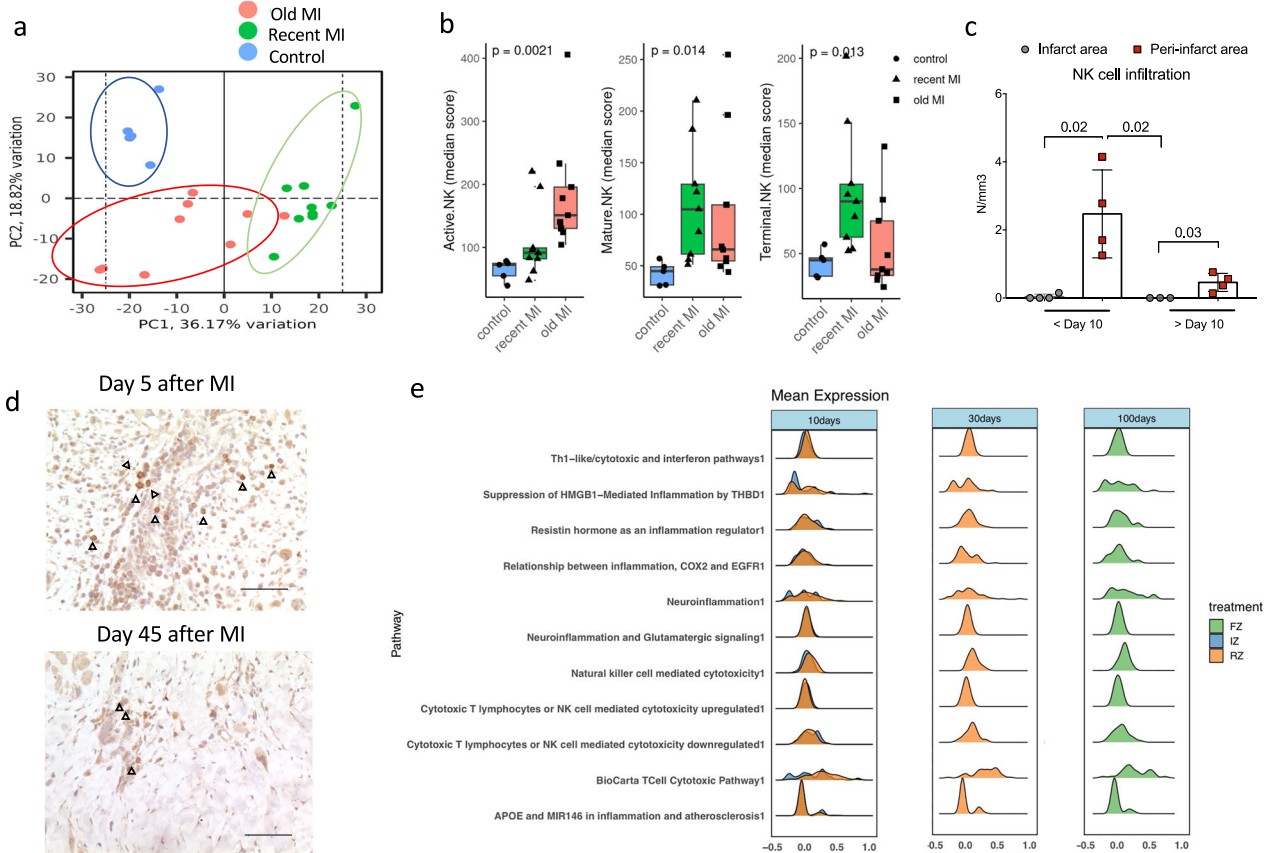

**Fig. 6 | NK cells in human myocardial infarction. a** Principal Component Analysis of the variable genes between groups ($n = 6$ control group, recent MI $n = 9$, old MI $n = 9$) based on Nanostring method. **b** NK subsets in the 3 groups of patients (box and whiskers: Median and 1st–3nd quartiles). **c** Quantification of NKp46[+] NK cells in human ischemic heart tissue at different timepoint after MI using immunohistochemistry ($N = 4$/group/timepoint). Data are presented as mean values ± SEM. **d** Representative pictures of NK staining cells (Black arrow) in human ischemic heart tissue at different timepoint after MI using immunohistochemistry (Representative of 4 stainings/group/timepoint). scale bar 20 μm. **e** Sc-RNAseq was performed on different areas of ischemic heart at different timepoints after MI and in the NK population, cytotoxicity or inflammation pathways were analyzed (Msigdb database) and scored (AddModuleScore). *P* values were calculated using Kruskal-Wallis test (**b**) or two-tailed Mann–Whitney test (**c**). MI myocardial infarction.

then induced in both groups of mice, and myeloid populations were analyzed at day 5 (Fig. 5n). NK cell counts in the BM were similar between $Nkp46^{Cre+/-}Csf2^{lox/lox}$ and $Nkp46^{Cre-}Csf2^{lox/lox}$ MI-operated mice (Supplementary Fig. 21). Notably, we observed a significant reduction in GMP count and monocytes in the BM of GM-CSF-deficient mice (Fig. 5o), associated with a decrease in neutrophils (Supplementary Fig. 22), monocytes, and CCR2[+] macrophages in ischemic heart tissue (Fig. 5p). Finally, we observed higher left ventricle ejection fraction in $Nkp46^{Cre+/-}Csf2^{lox/lox}$ mice when compared to littermate $Nkp46^{Cre-}Csf2^{lox/lox}$ controls (Fig. 5q).

## NK cells in human MI
To evaluate the relevance of our findings to human disease, we performed transcriptomic analysis of human heart biopsies using Nanostring Technology targeting a panel of 758 genes related to inflammatory pathways. We compared gene expression in healthy heart tissue with ischemic cardiac tissue at early stages (median time 7 (2–13) days) and late stages (median time 39 (3–108) months) after MI (Supplementary Fig. 23). The transcriptomic profile of the heart tissue was profoundly altered by ischemia, with clear distinctions in the principal component analysis between the 3 groups (Fig. 6a). We analyzed 3 NK cell subsets in heart tissue, previously characterized at the transcriptomic level[18]: active NK cells, characterized by up-regulation of inflammatory genes (*NR4A2, DUSP1, FOSB, FOS, JUN,*

and *JUNB)*; mature NK cells, expressing genes associated with cytotoxic activity (PRF1, GZMA, GZMB, GZMH); and terminal NK cells, which display a transcriptional profile similar to the mature NK subset but express the transcription factor ZEB2, indicative of terminal maturation. As depicted in Fig. 6b, Mature NK cell transcriptomic signature was significantly upregulated at early stages post-MI, whereas active NK cell transcriptomic signature was significantly upregulated at late stages post-MI. Immunostaining further confirmed the infiltration of NK cells in the heart tissues collected from patients with a recent coronary occlusion (Fig. 6c, d). In addition, we re-analyzed single-nucleus RNA sequencing data from human hearts (Kuppe et al. 2022). Samples were obtained from the necrotic area or the unaffected LV myocardium regions (remote zone) of patients with acute myocardial infarction, as well as from fibrotic zones of post-MI hearts at a later stage. By calculating module scores for cytotoxic and inflammatory pathways, we quantified the transcriptional activity of these pathways in individual NK cells (see "Methods"). This approach allowed us to identify subtle but significant shifts in pathway activity. As shown in Fig. 6e, we identified both pro-inflammatory and cytotoxic transcriptional signatures in NK cells infiltrating both ischemic and remote areas of the heart at day 10. By day 30, cytotoxic activity and inflammation were more pronounced in remote areas. Finally, by day 100, the fibrotic zone exhibited heightened cytotoxic activity. Finally, we investigated the chemokines/chemokines receptors in the human data

from Kuppe's work[19]. Cells were categorized into cardiomyocytes, non-cardiomyocytes (non-CM), NK cells and other immune cells. As shown in Supplementary Fig. 24, all major cell types exhibited high expression of chemokines CCL2 and CCL5. In contrast, CCR2 and CCR5 were predominantly expressed in immune cells and NK cells, respectively. Cardiomyocytes and non-CM cells showed minimal or no expression of these receptors, suggesting that immune and NK cells are the primary targets.

## Discussion

Previous studies have conclusively demonstrated that NK cells play a critical role in anti-viral and anti-tumor immune responses[16]. Additionally, NK cells participate in allograft vasculopathy[20], but their contribution to the inflammatory response following acute sterile damage, such as post-ischemic injury, remains poorly understood. Our study reveals a significant pathogenic role for NK cells in adverse cardiac remodeling following acute ischemia, driven by NK cell cytotoxic activity and their regulation of myeloid cell trafficking.

We observed that NK cells extensively infiltrate the ischemic myocardium in a murine model of MI. Mature cytotoxic NK cells were recruited within the first five days, while immature NK cells exhibiting an inflammatory signature were recruited at a later stage. This pattern of NK cell infiltration was observed mainly in the border zone of the ischemic heart tissue in both murine and human samples. Interestingly, the early acute infiltration of NK cells into the heart observed in our study corresponds with a reduction in circulating NK cells in humans following percutaneous coronary intervention[21].

Chemokine receptors involved in NK cell homing vary depending on the context (acute versus chronic), the type of disease, and the organ involved. For instance, in the context of stroke, NK cell homing to the ischemic brain is mediated by CX3CR1[22]. In our murine MI model, as well as in human ischemic heart tissue, scRNA-seq analysis revealed that CCR2, CCR5, and CXCR4 are the primary chemokine receptors expressed by NK cells, whereas CX3CR1 was not detected. NK cell recruitment to the ischemic heart was, at least partially, mediated by the CCR2 receptor, as NK cell numbers were significantly lower in $Ccr2^{-/-}$ mice. However, defining the specific contribution of CCR5 remains challenging, as CCR2 expression was found to be up-regulated in $Ccr5^{-/-}$ mice.

Our study employed both "loss of function" and "gain of function" methodologies, utilizing anti-NK1.1 depleting monoclonal antibodies and anti-NKG2A activating monoclonal antibodies, respectively. This approach demonstrated that NK cells contribute to harmful post-ischemic cardiac remodeling in both male and female mice. While the anti-NK1.1 mAb depleted both NK and NK T cells, the NK T cell population represents a much smaller population in the MI model, being approximately tenfold less abundant than NK cells. Consistent results were obtained from experiments using $Nkp46^{iCre+/-} R26R^{DTA}$ mice, a murine model with specific NK cell deficiency, reinforcing our conclusion about the pathogenic role of NK cells in experimental ischemic cardiac failure. Further support for our findings was provided by experiments using anti-NKG2A monoclonal antibody, which activates NKG2A in NK cells, but also CD8+ T cells. However, the aggravation of post-ischemic cardiac remodeling following anti-NKG2A treatment was also observed in CD8-depleted mice, excluding CD8+ T cells as confounding contributors to this effect. Our results hold significant translational implications, especially given that anti-NKG2A monoclonal antibodies are currently in development for oncology applications[23]. Our findings suggest that such therapies should be used cautiously in patients at high cardiovascular risk, as enhanced NK cell activation could exacerbate post-ischemic cardiac damage.

In vivo, we showed with several complementary approaches that NK cells significantly contribute to adverse ventricular remodeling through the induction of cardiac cell death. We confirmed in vitro that activated murine NK cells induced cardiomyocyte death, and that IL-2-activated human NK cells markedly impaired the function of 3D cardiac rings. In future experiments, it would be of interest to use NK cells derived from patients with MI; however this not an easy task. Similarly, NK cells can induce neuronal cell death in vitro, and their depletion using anti-NK1.1 antibodies improve brain infarction outcomes and neurological deficits in murine stroke models[22]. The protective effects of NK cell deactivation, exemplified by astragaloside from the herbal medicine *Astragalus mongholicus* Bunge, acting through STAT3-dependent pathways, underscore the therapeutic potential of regulating NK cell activity[24]. Mechanisms implicated in NK cell-mediated cell death include the Perforin/Granzyme complex, death receptor pathways such as TRAIL, and pro-inflammatory cytokines like Interferon-γ[25]. Our study specifically focused on Granzyme B, a serine protease released by NK cells that activate apoptosis pathways and trigger caspase activation, for three key reasons. First, a Granzyme B/Perforin transcriptomic signature was detected in infiltrating NK cells in the ischemic myocardium by sc-RNA seq analysis. Second, the temporal kinetics of *Gzmb mRNA* levels in the heart post-MI aligned with the timing of NK cell infiltration. Third, previous work from our group demonstrated the deleterious role of Granzyme B in cardiomyocyte apoptosis and cardiac tissue necrosis in experimental MI[6].

Fluorescent immunostainings revealed Granzyme B in NK cell cytoplasm and in the peri-infarct zone, indicating local degranulation. Granzyme B is a known cytotoxic molecule in auto-immune diseases including diabetes[26], and cardiovascular diseases such as stroke[22]. Our study demonstrated that Granzyme B mediates the pro-apoptotic effect of NK cells following MI. In vitro studies confirmed that activated NK cells induced cardiomyocyte apoptosis, which was significantly reduced in co-culture with $Gzmb^{-/-}$ NK cells. Furthermore, the detrimental cardiac effects of NK cell reconstitution in $NK^{KO}$ mice were alleviated when they were repopulated with $Gzmb^{-/-}$ NK cells. Therefore, our findings indicate that plasma levels of Granzyme B, previously recognized as a predictive factor in MI patients[6,27], could act as a biomarker to identify individuals who may benefit from NK-depleting therapies during acute MI. This mechanism-based approach should be assessed in future studies as anti-NK monoclonal antibody treatments become available.

NK cell depletion reduced fibrosis in the ischemic heart, whereas NK cell activation using anti-NKG2A monoclonal antibody aggravated fibrosis, as assessed by Masson's trichrome staining. These findings were consistent with the early post-MI quantification of *Col1a1* and *Col3a1* mRNA levels. We did not specifically investigate the mechanisms by which NK cells modulate fibrosis; however, it is unlikely to be mediated by Tnf or IFN-γ, as *Tnf* and *Tbet* (a key regulator of IFN-γ production) mRNA levels in the ischemic heart was similar between isotype-treated and anti-NKG2A treated mice. Nevertheless, further studies will be required to explore this mechanism in detail.

NK cell depletion led to a shift in the immune milieu of the ischemic heart towards a less inflammatory profile with a reduction in local pro-inflammatory cytokines and changes in macrophage transcriptomic profile. Such modifications may partly result from decreased cardiomyocyte apoptosis and subsequent reduction in DAMPs, which modulate macrophage polarization[28].

GM-CSF, a critical growth factor in leukocyte production, proliferation, differentiation, and survival[29,30] was identified as a key mediator produced by NK derived from BM during MI, regulating myelopoiesis. We showed that NK cell-specific deletion of *Csf2* results in decreased GMP and myeloid cells in the BM, leading to reduced myeloid cell infiltration into the ischemic heart. This myeloid phenotype in $Nkp46Cre^{+/-} Csf2^{lox/lox}$ mice mirrors the effects observed in global *Csf2* KO models of acute MI[31]. In the heart, GM-CSF from fibroblasts enhances local pro-inflammatory cytokine and chemokine production by macrophages[31]. While GM-CSF is difficult to detect in mice, its levels rise substantially in humans post-MI, correlating with LV remodeling[32,33].

NK trafficking, infarct size, and LV function were similar between *Ncr1*[+/+] and *Ncr1*[gfp/gfp] mice, ruling out a specific role of the NKp46 receptor in NK cell pathogenic activity in the MI model. We can speculate that NK cell could be activated through NKG2D for several reasons. First, we have shown that NKG2D is expressed by infiltrating NK cells in ischemic cardiac tissue. Next, others have reported that the deletion of *Klrc1* gene (encoding NKG2D) protected against ischemic heart failure in murine models[34]. Moreover, it was reported that Rae-1ε, a ligand for NKG2D, is expressed by cardiomyocytes in ischemic conditions[34]. NK cells may be activated either within the ischemic cardiac tissue or directly in the bone marrow by circulating factors. Parabiosis experiments have demonstrated that factors released during MI can modulate hematopoiesis[35]. Other studies suggested that DAMPs and cardiomyocyte-derived extracellular vesicles may also contribute to this process[36]. A comprehensive investigation of these mechanisms will require dedicated future studies.

Our study highlights a global pathogenic role for NK cells in the immune-inflammatory responses contributing to tissue damage following acute MI. However, we cannot rule out a protective role for specific NK cell subsets at later stages, as previous studies have reported pro-angiogenic activities of NK cells[37]. We further showed that NK cells rapidly migrate to ischemic heart tissue in a CCR2-dependent manner, where they release Granzyme B, a cytotoxic protease that induces cardiomyocyte death and triggers a sterile pro-inflammatory response. Consequently, depletion of NK cells substantially reduces cardiomyocyte death, myocardial inflammation, and infarct size, ultimately leading to improved myocardial function. The presence of an NK cell-specific cellular and transcriptomic signature in human ischemic heart tissue underscores the clinical relevance of our findings. These insights indicate that targeting NK cells may be a promising therapeutic strategy for acute MI. In particular, developing and administering humanized NK cell-depleting antibodies during the acute phase of MI could attenuate cardiac damage and preserve cardiac function, offering a novel approach for future clinical interventions in ischemic cardiovascular disease.

## Methods

### Mice models

All experiments were conducted according to the French veterinary guidelines and those formulated by the European Community for experimental animal use and were approved by the Institut National de la Santé et de la Recherche Médicale. Permission was granted to perform animal experiments by the appropriate committee (the ethical committee CEEA34 Université de Paris, APAFIS #29531-2021020416255418v2). All the animals were maintained under identical standard conditions (housing, regular care, and normal chow). Mice were maintained in isolated ventilated cages under specific pathogen-free conditions. Experimental/control animals were bred separately.

### Myocardial infarction

All mice used in this study were of the C57Bl/6 J strain. Specifically, C57BL/6 J mice (sourced from Janvier, France, Stock designation: *C57BL/6JRj*), *Gzmb*[−/−] mice (from Jackson Laboratory, USA, Stock No. 002248), *Nkp46*[iCre+/−] *R26R*[DTA] mice (provided by E. Vivier's lab in Marseille, France[38]), *Ncr1*[gfp/gfp] mice (from O. Mandelboim's lab in Jerusalem, Israel[15]), and *Csf2*[Lox/lox] mice (from I.P. Wicks at WEHI in Melbourne, Australia[17]) were used. *Ccr2*[−/−] mice were obtained from Jackson Laboratory (USA) and *Apoe*[−/−]*Ccr5*[−/−] (named *Ccr5*[−/−] in the manuscript) came from Y. Döring's Lab[39] (Switzerland). All mice were aged 9–10 weeks at the time of myocardial infarction (MI) induction. All the experiments were performed on 8–10-week old male mice. The protective effect of NK depletion using anti-NK1.1 mAb was also validated on female mice. No sample size calculation were performed.

The MI was induced by ligating the left anterior descending coronary artery, as described by Santos-Zas et al.[6]. Mice were anesthetized with an intraperitoneal injection of ketamine (100 mg/kg) and xylazine (10 mg/kg), intubated, and ventilated using a small animal respirator (SAR830A/P, CWE). After shaving the chest wall, a thoracotomy was performed in the fourth left intercostal space to expose the left ventricle. The pericardial sac was removed, and the left anterior descending artery was permanently ligated with a 7/0 non-absorbable monofilament suture (Peters Surgical, France) at the point where it emerges under the left atrium. A successful coronary occlusion was confirmed by significant color changes in the ischemic area. The thoracotomy was closed with 6/0 non-absorbable monofilament sutures (Peters Surgical, France). Sham-operated controls underwent the same procedure without tying the ligature. After the surgery, the endotracheal tube was removed when spontaneous breathing resumed, and the mice were placed on a warming pad at 37 °C until fully awake. To ensure compliance with ARRIVE guidelines and our institutional ethical board, animal welfare was monitored via a clinical scoring system (0–3) evaluating body weight, physical appearance, behavior, and wound healing. Mandatory euthanasia was performed via cervical dislocation under isoflurane sedation for any single category score of 3 or a cumulative score of 6, while intermediate scores (3–5) were managed with buprenorphine analgesia and systematic re-evaluation within 24–72 h.

To study NK cell recruitment and activation in the ischemic heart post-MI, C57BL/6 J wild type mice underwent permanent coronary ligation. Sham-operated C57BL/6 J wild type mice served as controls, undergoing the same procedure without the coronary ligation. Mice were euthanized by cervical dislocation by trained and experienced staff, at various time points for flow cytometry and histological analysis. To evaluate the role of NK cells in post-ischemic cardiac remodeling, C57BL/6 J Wild type mice were injected i.p. with a depleting mouse anti-mouse NK1.1 monoclonal antibody (200 μg/mouse, Biox-Cell BE0036 clone PK136) or with a mouse IgG2a isotype control (200 μg/mouse BioxCell BE0085 clone 2A3) 1 h after coronary artery ligation. In order to maintain cell depletion for an extended period, mice were re-injected at day 7 and day 14 after MI. Animals were euthanized 24 h, 3, 5, 7, 14 and 21 days after surgery for flow cytometry and histological analysis. To activate in vivo NK cells, C57BL/6 J wild type mice were injected i.p. with a neutralizing rat anti-mouse NKG2A/C/E monoclonal antibody (200 μg/mouse, BioXCell BE0321, clone 20D5) or with a rat IgG2a isotype control (200 μg/mouse BioxCell BE0085 clone 2A3) 1 h after coronary artery ligation. In order to maintain cell activation for an extended period, mice were re-injected once a day every day until day 7 after MI. To explore the role of NKG2D in the context of MI, isotype- or anti-NK1.1-treated C57BL/6 J mice were injected i.p. with a neutralizing Armenian hamster anti-mouse anti-NKG2D monoclonal antibody (200μg/mouse, BioxCell BE0111, Clone HMG2D).

Next, 8 week-old NK deficient *Nkp46*[iCre+/−] *R26R*[DTA] mice were used as recipients for NK cell repopulation studies 3 days prior MI. Purified NK cells, obtained from C57BL/6 J WT mice or *Gzmb*[−/−] mice, were resuspended in phosphate-buffered saline (2 × 10⁶ cell/ 100 μl/ mouse) and transferred to *NK deficient* mice by i.v injection 3 days before MI. Repopulation was checked at day 3 following MI in the heart.

### Echocardiographic measurements

Transthoracic echocardiography was performed at 21 days after surgery using an echocardiograph (VEVO2100, Visualsonics) and an 18 to 38-MHz transducer (MS400). The investigator was blinded to group assignment. Animals were anesthetized by isoflurane inhalation. Two-dimensional parasternal long-axis views of the left ventricle were obtained for guided M-mode measurements of the LV internal diameter at end diastole (LVDD) and end systole (LVDS), as well as the interventricular septal wall thickness, and posterior wall thickness at

the same points. Ejection fractional percentage (% EF) was calculated using Simpson method.

## NK cell purification and transplantation

NK cells were isolated from C57BL/6 J, $Gzmb^{-/-}$ spleens and purified using a mouse NK cell isolation kit (130-115-818 Miltenyi Biotec; Paris, France) according to the manufacturer's protocol. In brief, NK cells were negatively selected using a cocktail of antibody-coated magnetic beads followed by cell separation using LS magnetic columns (Miltenyi Biotec; Paris, France), yielding NK cells with >90% purity. Then cells were: (a) intravenously injected 3 days prior MI in $Nkp46^{iCre+/-}$ $R26R^{DTA}$ mice (b) cultured for in vitro experiments.

## NK cell culture

in vitro. NK cells were isolated as described above and cultured in activation medium containing RPMI-1640 supplemented with 10% fetal bovine serum, 100 units/ml penicillin, 100 units/ml streptomycin (Gibco, Thermofisher) 5 ng/ml IL15 (R&D Systems; Abingdon, United Kingdom). To confirm GM-CSF deficiency in $Nkp46^{Cre+/-} Csf2^{lox/lox}$ mice, purified NK cells were cultured at 37 °C with 5% CO2 in activation medium containing RPMI-1640 supplemented with 10% fetal bovine serum, 100 units/ml penicillin, 100 units/ml streptomycin (Gibco, Thermofisher), rIL-15 (10 ng/ml), rIL-18 (50 ng/ml), and rIL-12 (50 pg/ml, R&D Sytems). After 48 h, the supernatant was recovered and GM-CSF levels were measured by ELISA (MGM00, R&D Systems), following manufacturers protocol.

## HL1 cardiomyocytes culture

Cells were seeded into 24-well plates (coated with gelatin-fibronectin (Sigma-Aldrich) overnight, 37 °C) at a density of 20 ×103 cells/ml/well and maintained during 2 days in Growth Medium containing Claycomb Medium (Sigma-Aldrich) supplemented with 10% fetal bovine serum, 100 units/ml penicillin, 100 units/ml streptomycin (Gibco, Thermofisher), 1% norepinephrine 10 mM (Sigma-Aldrich) and 1% L-Glutamine 100X (Sigma-Aldrich).

## Co-culture and apoptosis experiment

Activated (rIL-15, 10 ng/ml) and non-activated NK cells were co-cultured directly with HL1 cardiomyocytes in different ratios (1:1, 1:5, and 1:10) with respect to cardiomyocyte count. Following co-culture for 24 h, NK cells were removed and HL1 cardiomyocytes were labeled with the addition of 5 µM Caspase-3/7 Green Apoptosis Reagent (Incucyte, Welwyn Garden, United Kingdom) to the growth medium. After 30 min at 37 °C in dark, apoptosis in real time was evaluated with a microscope Nikon Eclipse, Ti. Time-lapse images were captured at 10 min intervals. To evaluate GM-CSF production by NK cells ex vivo, cells were stimulated with rIL-12 (50 microg/ml), rIL-15 (10 ng/ml), and rIL-18 (50 ng/ml)[17].

## 3D cardiac rings generation and recording

3D cardiac rings were generated as previously described[40]. The tissues were generated on a commercial SmartHeart® 96-well plate (4DCell, Montreuil, France), which contains a molded hydrogel in each well that induces the seeded cells to self-organize into ring-shaped tissues. On day 22 of differentiation, human cardiomyocytes derived from induced pluripotent stem cells (hiPSC-CMs) were dissociated and mixed with human dermal fibroblasts at a ratio of 1:3 (hiPSC-CM:fibroblast). Nine days after their generation, the tissues were imaged at baseline using brightfield microscopy with a high-speed CCD camera (PLD672MU, Pixelink) mounted on Primovert microscope (Zeiss) at 10X magnification. The tissues were then incubated for 12 h before recording with either non-activated or IL-2-activated human Natural Killer cells, at hiPSC-CMs:NK ratios of 1:1 and 1:5. Automated video analysis was performed using a Matlab script to recover contractility parameters such as contraction frequency, contraction stress and the maximum contraction and relaxation speeds. The protocol for human iPS was approved by the local institutional ethics committee (Comité de Protection des Personnes Ile de France XI IRB number 11-015). Human NK cells were isolated from blood donors (Collaboration *Etablissement Français du sang*) using kits (Miltenyi biotech, Paris, France), (catalog #130-092-657) and next were activated or not in vitro overnight with IL-2r (100 IU/mL) (IRB 2022-2026-046 Comité de Coordination de la Protection des Sujets et des Locaux Imagine). Next, cells were washed and co-cultured with 3D cardiac rings at different ration during 12 h. All the volunteers and blood donors gave informed consents.

## Histopathological and immunofluorescence analyses

Cardiac healing following myocardial infarction was assessed at different time points. Hearts were excised, rinsed in PBS and frozen in liquid nitrogen. For tissue morphology analysis after MI, hearts were cut along their length into 7 µm thick cardiac muscle cryosections (CM 3050S, Leica). Serial sections were mounted on microscope slides ($n = 10$), each section being spaced from the next one by 400 µm. In this way, 10 sections / hearts allowed entire heart tissue analysis. Masson's trichrome and Sirius Red stainings were performed for infarct size and interstitial fibrosis evaluation, respectively. Infarct size was calculated as a percentage of infarct area to total LV surface. The collagen volume fraction was calculated as the ratio of the total area of interstitial fibrosis to the myocyte area in the entire visual field of the section.

Heart sections for immunofluorescence analysis were fixed with paraformaldehyde 4%, permeabilized using 0.2% Triton X100 in Phosphate Buffer Solution (PBS) 30 min at room temperature, blocked with PBS-T (0.2% Triton X100, 10% goat serum, 0,2% BSA in PBS) for 1 h, and incubated with primary antibodies diluted in PBS-T overnight at 4 °C: NK cell infiltration was detected using an anti-NKp46 antibody (Abcam ab233558, 1:200); Granzyme B detection in ischemic heart tissue was performed using an anti-Granzyme B antibody (R&D Systems AF1865, 1:100); Sections were washed with PBS and incubated with a mixture of appropriate secondary antibodies for 1 h at room temperature: Cyanine 3 Goat anti-rat and Cyanine 5 Donkey anti-goat, respectively (Jackson Immunoresearch, 1:250). 4',6-diamidino-2-phenylindole (DAPI) was used to counterstain cell nuclei.

To evaluate cell apoptosis (day 3 post-MI), immunofluorescence analyses were performed using a TUNEL assay kit (Roche Diagnostis, Meylan, France) according to manufacturer's instructions. The digital images of immunofluorescence were acquired with a Zeiss Axioimager Z2 Apotome. Six microscopic fields in both sides of peri-infarct area from each heart were examined using ImageJ64.

## TTC staining

2,3,5-Triphenyltetrazolium chloride was used to evaluate infarct size at early time points (3 days after MI). TTC staining identifies metabolically active tissue and is helpful to measure tissue viability. Hearts were removed and sectioned into 2 mm thick slices in a semi frozen state. The slices were then immersed 40 min into 1% TTC solution at 37 °C. Once the color had been established, slices were fixed in 4% PAF overnight at room temperature. Slices were photographed and infarcted heart tissue areas were calculated using ImageJ64 blindly.

## Ex vivo reflectance epifluorescence imaging

Mice were anaesthetized with isoflurane and received intravenously 150 µL of a fluorescent imaging probe MMPsense 680 (NEV 10126, PerkinElmer) 24 h before euthanasia. This agent is optically silent in its un-activated state and becomes highly fluorescent following activation by MMPs including MMP-2, -3, -9, and -13. Images of ischemic hearts were acquired using a fluorescence molecular imaging system (FMT 2500TM, VisEn Medical)[41].

## Cell suspension preparation for flow cytometry

Mice were euthanized on days 1, 3, 5, 7, 14, and 21 post myocardial infarction. Peripheral blood was drawn via inferior vena cava puncture using a syringe primed with heparin solution. For blood staining, erythrocytes were lysed using BD FACS lysis solution (BD Biosciences). Spleens were surgically removed and dissociated obtaining a single cell suspension that was filtered through 40-μm nylon mesh (BD Biosciences). Cell suspensions were centrifuged at $400 \times g$ for 15 min at 4 °C. Red blood cells were removed using a red blood lysis buffer (Sigma-Aldrich) and splenocytes were washed with PBS supplemented with 3% FBS. Left ventricle of infarcted and control mice were harvested, minced with fine scissors, digested using a solution of collagenase II and IV (both at 10 mg/mL) for 30 min at 37 °C. Upon digestion, tissue is gently passed through a 100-μm nylon mesh (BD Biosciences) followed by a 40-μm nylon mesh (BD Biosciences) and centrifuged at $400 \times g$ for 15 min. Red blood cells were removed by incubation in red blood cell lysis buffer (Sigma-Aldrich) and single cell suspension were washed with PBS supplemented with 3% FBS.

## Flow cytometry

General characteristics of antibodies are summarized in the supplementary Table 1. Surface stainings were performed before permeabilization when intracellular labeling was necessary. Cells were incubated in Fixation/Permeabilization buffer (eBiosciences) 45 min, washed with permeabilization buffer (eBiosciences) and stained with the appropriate marker for intracellular staining. Forward scatter (FSC) and side scatter (SSC) parameters were used to gate live cells excluding red blood cells, debris, and cell aggregates in total splenocytes. Cells were analyzed BD LSR FORTESSA flow cytometer (BD Biosciences).

## Quantitative real-time PCR

Quantitative real-time PCR was performed on a Step-one Plus (Applied Biosystems) qPCR machine. *Gapdh* was used to normalize gene expression. The primer sequences are described below:

*Gapdh* Forward 5′-CGTCCCGTAGACAAAATGGTGAA-3′, Reverse 5′-GCCGTGAGTGGAGTCATACTGGAACA-3′; *Gzmb* Forward 5′-GTGCGGGGGACCCAAAGACCAAAC-3′, Reverse: 5′-GCACGTGGAGGTGAACCATCCTTATAT-3′; *Il1β* Forward 5′-GAAGAGCCCATCCTCTGTGA-3′, Reverse 5′-GGGTGTGCCGTCTTTCATTA-3′; *Il6*

Forward 5′-TGACAACCACGGCCTTCCCTA-3′, Reverse: 5′-TCAGAATTGCCATTGCACAACTCTT-3′; *Il10*, Forward 5′-ACTTCCCAGTCGGCCAGAGCCACAT-3′, Reverse: 5′-GATGACAGCGCCTCAGCCGCATCCT-3′; *Tnf* Forward 5′-GATGGGGGGGCTTCCAGAACT-3′, Reverse 5′-GATGGGGGGCTTCCAGAACT-3′; *Mmp9* Forward 5′-GCGTCATTCGCGTGGATAAGGAGT-3′, Reverse 5′-GTAGCCCACGTCGTCCACCTGGTT-3′; *Gmcsf* Forward 5′-TACGGGGCAATTTCACCAAACTC-3′, Reverse 5′-ATGAAATCCGCATAGGTGGTAACTTGT-3′; *Ccl2* Forward 5′-ATGCTTCTGGGCCTGCTGTTCA-3′, Reverse 5′-GAGTGGGGCGTTAACTGCATCTG-3′; *Mmp2* Forward 5′-CCGAGACCGCTATGTCCACTGT-3′, Reverse 5′-CCGGTCATCATCGTAGTTGGTTGT-3′; *Gzma*, Forward 5′-GTGCAGGGGACCTCCGTGGTGGAAAG-3′, Reverse 5′-GCCATCGGCGATCTCCACACTTCTCT-3′; *Ncr1* Forward 5′-CAGCCTTGCCACCTACCGACCCTAC-3′, Reverse 5′-GATCCCAGAAGGCGGAGTCCTTTT-3′; *Csf2* Forward: 5′ AACCTCCTGGATGACATGCCTG 3′, Reverse: 5′ AAATTGCCCCGTAGACCCTGCT 3′; *Col1a1* Forward 5′-TGCCCTCCTGACGCATGGCCAAGA-3′, Reverse 5′-CCTCGGGTTTCCACGTCTCACCATT-3′; *Col3a1* Forward 5′-CACCCCGAACTCAAGAGTGGAGAA-3′, Reverse 5′-ATGGGGCTGGCATTTATGCATGTT-3′

## Single-cell RNA-sequencing (mouse)

Gene expression patterns in NK cells were analyzed in previously published single-cell RNA-seq data of total CD45[+] cells from control and infarcted C57BL6/J mouse hearts (day 5 after MI) from reference [11]. Briefly, cells corresponding to T and NK cells as identified in ref. [11] were extracted and reclustered in Seurat v4[42], with batch correction of

separate scRNA-seq libraries performed using harmony[43]. Clustering was performed using 30 principal components and a 0.6 resolution. NK cells were identified based on the expression of surface NK1.1 as measured by cellular indexing of transcriptomes and epitopes by sequencing (CITE-seq)[10].

C57BL/6 J wild type mice underwent permanent coronary ligation and were injected i.p. with a depleting mouse anti-mouse NK1.1 monoclonal antibody (200microg/mouse, BioxCell BE0036 clone PK136) or with isotype 1 h after coronary artery ligation. Five days after MI, the mice were injected i.v. with 2.5 μg anti-CD45.2 APC (clone 104, Biolegend, cat. #109814) under isoflurane anesthesia, 5 min before being killed. The heart was perfused with PBS, excised, and the infarct, peri-infarct area, and adjacent viable myocardium collected, weighed and digested for 1 h at 37 °C under agitation in RPMI containing 450U/ml collagenase I (Sigma-Aldrich, #C0130), 125U/ml Collagenase XI (Sigma-Aldrich, C7657), 60U/ml Hyaluronidase (Sigma-Aldrich, H3506), 60U/ml DNAse (Roche #11284932001). After digestion, the cell suspensions were passed through a 70 μm cell strainer and washed twice in MACS buffer (PBS + 0.5% BSA + 2 mM EDTA) and incubated for 5 min on ice with TruStain FcX™ (anti-mouse CD16/32, Biolegend #101320, 10 μg/ml). Afterwards, cells were incubated with mouse CD45 microbeads (Miltenyi, 130-052-301, 1:10 final concentration), and anti-mouse CD45.2-AlexaFluor488. After 10 min incubation at 4 °C, TotalSeq-A hashtag antibodies (Biolegend, see table for CITE-seq panels) were added as follows: hashtag 1 to 5 controls; hashtags 6 to 10: NK-depleted. Cells were incubated for an additional 15 min at 4 °C, washed twice in MACS buffer, pooled, and magnetic selection was performed using LS Columns (Miltenyi (# 130-042-401)) and Mid-iMACS separators (Miltenyi # 130-042-302). Sorted CD45[+] cells were washed twice with PBS containing 0.04% BSA (Sigma-Aldrich A1595) and incubated for 25 min at 4 °C in PBS/0.04% FCS containing Fixable Viability Dye eFluor™ 780 (ThermoFisher #65-0865-14, 1:1000) and anti-mouse TotalSeqA-Antibodies (see CITE-seq panels summary below). Cells were washed with PBS/0.04% FCS, resuspended and sorted using a BD FACS Aria III with a 100 μm nozzle. Cells were resuspended, counted (final concentration 1650 cells/μl) and loaded in the 10x Genomics Chromium using Chromium Next GEM Single Cell 3′ High Throughput (HT) Reagent Kits v3.1.

Samples were prepared for CITE-Seq/Hashing according to the manual until the cDNA amplification step in which 1 μl (0.2 μM) ADT PCR additive primer to capture Antibody-derived tags (ADTs) and 1 μl (0.1 μM) HTO PCR additive primer to capture Hashtag oligos (HTOs) were added. After cDNA amplification 60 μl (0.6x) SPRI beads (Beckman Coulter) were added to separate the supernatant fraction that contains the ADT/HTO-derived cDNAs ( <180 bp) and the bead fraction that contains the mRNA-derived cDNAs ( >300 bp). The mRNA-derived cDNAs (bead fraction) were processed following the standard 10x Genomics protocol. The ADT/HTO-derived cDNAs (supernatant fraction) were purified twice with 2x SPRI. After the two purifications, half the amount of the eluted cDNAs were amplified for ADTs and half for hashtags. In the same reactions, the ADTs were indexed using TruSeq Small RNA primers and the hashtags using modified TruSeq DNA primers. The libraries were purified once more with 1.6x SPRI (160 μl). All libraries were quantified by QubitTM 3.0 Fluometer (ThermoFisher) and quality was checked using 2100 Bioanalyzer with High Sensitivity DNA kit (Agilent). Sequencing was performed with S1 or S2 100 bp flowcell with Novaseq 6000 platform (Illumina) and the reads for CITE-Seq/Hashing sample were allocated as follows: 5% for the hashtags, 10% for the ADTs and 85% for the mRNAs. 10x Genomics data, including HTO and ADT libraries, was demultiplexed using Cell Ranger software (version 7.0.1). Mouse GRCm38 (mm10) reference genome was used for the alignment and counting steps. To evaluate the expression of the cell surface proteins alongside the transcriptome level in our main data set, --feature-ref flag of the Cell Ranger software was used which creates a matrix that contains gene expression counts

alongside the expression of cell surface proteins. The gene-barcode matrix obtained from Cell Ranger was further analyzed using R 4.5.1 and Seurat 5.3.0.[44]. Clustering was performed using 20 principal components and a 0.4 resolution. Cells were identified based on their top10 DEG and classical markers. Using hashing identification, samples containing less than 2000 cells were excluded, further analysis were conducted with 6 samples in total (3 control and 3 depleted). Myeloid cells (Monocytes, macrophages and neutrophils) were extracted for further analysis. Pseudo bulk analysis were run for each cell population to identified DEG across control and depleted group with DESeq2. Among them, *Cd36*, *Hdac9* and *P2rx7* were identified, and their global expressions were assessed in Monocytes, Mac Selenop, Mac *Spp1* and Mac resident, and presented with violin plots, where statistics (*t* test) were calculated for each cell type between control and depleted group with SeuratExtend. *Tnf* expression was assessed in Monocytes, Mac *Selenop*, Mac *Spp1* and Mac resident, Neutro SiglecF and Neutro *Lcn2*, and presented with violin plots, where statistics (t-test) were calculated for each cell type between control and depleted group with SeuratExtend.[45].

### Transcriptomic analysis (nanostring)

RNA extraction. Up to three 20-μm thick sections were collected from FFPE tissue blocks to obtain 100 ng of RNA required for analyses. RNA was isolated and extracted using the tissue RNEasy FFPE kit #73504 (Qiagen, Hilden, Germany). The concentration and quality of isolated RNA were assessed using NanoDrop 2000 spectrophotometer (Thermo Fisher Scientific Inc., Waltham, MA, USA).

RNA sequencing. Total RNA from each sample was hybridized with the nCounter® Human Banff Organ Transplant gene panel (B-HOT, NanoString Technologies, Seattle, WA, USA). This panel evaluates mRNA expression of 758 target genes and 12 selected housekeeping genes for data normalization. Internal quality reference for each assay was ensured by including a Panel Standard in each run: a pool of synthetic DNA oligonucleotides corresponding to the endogenous probes, allowing normalization of user, instrument, and lot-to-lot variations.

Quality control and data normalization. Raw counts of gene expression data were analyzed with the nSolver Analysis Software (version 4.0.70) and subjected to normalization using housekeeping genes. Background correction was applied, and the means of the supplied controls and housekeeping genes were used to normalize the measured expression values.

To evaluate the accuracy and quality of the gene expression input data, we applied quality control (QC) metrics (nSolver documentation) in all the samples included in the study. The QC assays refer to imaging, binding density, linearity of the positive controls, and limit of detection (LOD). If one or more of the four QC evaluations flagged an issue, the sample was removed from downstream analysis. Quantity and quality of isolated RNA were adequate in all cases and all the samples met the QC metrics.

All the samples passed the QC metrics which were defined as binding density >0.05 and <2.25 probes/μm[2], % fields of view >0.75, positive control linearity >0.95, and sufficient assay efficiency. Per-sample LOD was defined as the mean of the counts of negative control probes for each sample and was ensured by using internal quality control reference probes. Raw gene expression data were normalized using the Remove Unwanted Variation (RUV) approach following quality control and housekeeping assessment.

### Single-cell RNA-sequencing (Human)

As described in the original work[19], using single-nucleus RNA sequencing, we analyzed human heart data including thirty-one samples from twenty-three different individual hearts (Kuppe et al. 2022). The samples were taken from the necrotic area (ischemic zone from 12 samples), the unaffected LV myocardium regions (remote zone from 6 samples), patients with acute myocardial infarction, and the human heart specimen from the late stage of myocardial infarction (fibrotic zone from 6 samples). These areas were identified based on discrete cardiac myocardial time points from the patients. There were 191,795 cells in all over 29,126 features in the data. We subtracted the data on natural killer cells because we were primarily interested in immune cells, and the result was 696 cells. Based on area, the data was further separated into 10, 30, and 100 days. We divided the patients into several time periods based on distinct zones of myocardial infarction in order to ascertain the function of NK cells in inflammation and cytotoxicity. 10, 30, and 100 days were the three distinct time periods into which the data was split comprising of 8, 1, and 2 patient samples, respectively. Next, we gathered the pathways linked to cytotoxicity or inflammation and used the AddModuleScore tool to get the scores for each cell. The scores for each pathway that displayed the gene expression trend linked with them were then obtained.

### Immunostaining of human ischemic heart tissue

Normal heart tissue was obtained from Creative Bioarray company (United States). The pathological human cardiac tissue samples were collected in the IRB-approved CVMR biobank in our institution (IRB-approval CVMR-PRB HEGP authorization Comité de Protection des Personnes Ile de France 2 2016-13-09 MS2). All the patients gave informed consents for the surgical procedure. All the patients had left ventricle assist device (HeartMate II or HeartWare) implantation for severe cardiac failure. For implantation of the device the apex of the left ventricle was open and tissue samples were excised. Tissues were fixed in formalin and paraffin embedded. The tissue sections were stained with H&E for pathological diagnostic procedure allowing dating of myocardial infarction in addition to clinical dating. On remnant sections immunohistochemistry was performed with anti-NKp46 (clone 8E5B, Innate Pharma).

### Statistical analysis

Values are expressed as mean ± s.e.m. Differences between values were examined using the non-parametric Mann–Whitney test or Kruskal-Wallis test and were considered significant at $P < 0.05$. Shapiro-Wilk test was used for assessment of normality. All statistical tests were two-sided and performed using SAS software version 9.4.

### Reporting summary

Further information on research design is available in the Nature Portfolio Reporting Summary linked to this article.

## Data availability

The authors declare that the data supporting the findings of this study are available within the paper and its supplementary information files. Single-cell RNA-sequencing data of mouse heart generated for this report has been deposited in Gene Expression Omnibus (GSE317004). Transcriptomic data of human heart tissue generated using Nanostring technology has been deposited in Gene Expression Omnibus (GSE325257). As described in the original work[19] human data are available at cellxgene https://cellxgene.cziscience.com/collections/8191c283-0816-424b-9b61-c3e1d6258a77 and at the Zenodo data archive (https://zenodo.org/record/6578047). Raw data generated by Kramann's group (Institute of Experimental Medicine and Systems Biology, Aachen University, Germany) by CellRanger and SpaceRanger pipelines are available through the Human Cell Atlas Data Portal at https://data.humancellatlas.org/explore/projects/e9f36305-d857-44a3-93f0-df4e6007dc97 and at the Zenodo data archive (https://zenodo.org/record/6578553, https://zenodo.org/record/6578617 and https://zenodo.org/record/6580069). All data are included in the Supplementary Information or available from the authors, as are unique reagents used in this Article. The raw numbers for charts and

graphs are available in the Source Data file whenever possible. Source data are provided with this paper.

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

## Acknowledgements

This work was supported by Inserm, ANR (ANR-18-CE14-0009–TREKING to H.A.O., V.D), la fondation pour la recherche médicale (R.C.), la Fondation Lefoulon-Delalande (Y.S.Z.), the University Research Federation against heart failure (FHU2019, PREVENT_Heart Failure, to J.S.H and O.D.), and the Deutsche Forschungsgemeinschaft (SFB1123-A1)(Y.D. and C.W). SFB1525 project number 453989101 to A.-E.S and C.C. IW was supported by an Australian National Health and Medical Research Council Program Grant (1113577). We thank the PARCC iPS$_{ORG}$ core facility of Paris Cardiovascular Research Center for their help for human cardiac organoid experiments and the Single-Cell Center Würzburg, Alexander M. Leipold, and Tobias Krammer (Helmholtz-Center for Infection Research (HZI), Helmholtz Institute for RNA-based Infection Research (HIRI), Würzburg, Germany) for assistance in generating scRNA-seq data.

## Author contributions

R.C., V.D., R.AR., S.M, I.S.Z., R.B., S.T., T.G., L.C. performed the experiments, acquired and interpreted the data. R.C. and V.D. performed and interpreted the ultrasound studies. M.K. and Y.D. performed flow analysis on *Ccr5*[-/-] mice. A.L., O.D. and M.L participate in nanostring experiments. M.D., S.M and S.H. performed bioinformatics analysis on human transcriptomic data. J-S.S. contributed to data acquisition and analysis. C.J., J.S.H, S.N. and O.H. performed experiments with human cardiac organoids. P.B. analyzed and interpreted staining on human heart tissues. A.T., S.T. and J-S.S. contributed to study design and data interpretation. A-E.S., M.P., A.Z., E.T.S., G.R. and C.C. conducted scRNA-seq analysis. E.V., I.W. and C.W. reviewed the manuscript for intellectual content. E.V. and I.W. helped with GM-CSF experiments. R.C., V.D., R.AR. and H.A.O. designed the experiments, analyzed and interpreted the data, and wrote the manuscript.

## Competing interests

H.A.-O. and V.D. applied a patent on the protective effect of NK depletion blockade after acute myocardial infarction. The other authors report no conflicts. H.A.-O. is co-founder of POLYGON therapeutics, a biotech developing monoclonal antibodies against cytotoxic cells. E.V. is co-founder of Innate Pharma, a biotech developing antibodies targeting NK cells in Cancer field. The remaining authors declare no competing interests.

## Additional information

[1]Université Paris Cité, INSERM U970, Paris Cardiovascular Research Center, Paris, France. [2]Institute of Experimental Medicine and Systems Biology, RWTH Aachen University, Aachen, Germany. [3]Laboratorio de Endocrinología Celular, Área de Endocrinología Molecular y Cellular Instituto de Investigación Sanitaria de Santiago, Complejo Hospitalario Universitario de Santiago (CHUS), Santiago de Compostela, Spain. [4]Clinique Saint Gatien Alliance (NCT +), Saint-Cyr-sur-Loire, France, INSERM U1151, Necker Enfants Malades (INEM), IMMEDIAB Laboratory, Paris, France. [5]Institut Imagine, INSERM U1163, CNRS ERL 8254, Sorbonne Paris-Cité, Laboratoire d'Excellence GR-Ex, Université Paris Cité, Paris, France. [6]Institute for Cardiovascular Prevention (IPEK), Ludwig-Maximilians-University Munich (LMU), Munich, Germany. [7]German Centre for Cardiovascular Research (DZHK), Partner Site Munich Heart Alliance, Munich, Germany. [8]Division of Angiology, Swiss Cardiovascular Center, Inselspital, Bern University Hospital, University of Bern, Bern, Switzerland. [9]Department for BioMedical Research (DBMR), University of Bern, Bern, Switzerland. [10]Department of Biochemistry, Cardiovascular Research Institute Maastricht (CARIM), Maastricht University Medical Centre, Maastricht, The Netherlands. [11]Munich Cluster for Systems Neurology (SyNergy), Munich, Germany. [12]CIC1418 and DMU CARTE, AP-HP, Hôpital Européen Georges-Pompidou, Paris, France. [13]Institute of Experimental Biomedicine, University Hospital Würzburg, Würzburg, Germany. [14]Helmholtz Institute for RNA-based Infection Research (HIRI), Helmholtz-Center for Infection Research (HZI), Würzburg, Germany. [15]Institute of Molecular Infection Biology (IMIB), University of Würzburg, Würzburg, Germany. [16]Walter and Eliza Hall Institute of Medical Research and Department of Medical Biology, University of Melbourne, Parkville, Melbourne, VIC, Australia. [17]Aix Marseille Université, CNRS, INSERM, Centre d'Immunologie de Marseille-Luminy, Marseille, France. [18]Sorbonne Université, Service de médecine intensive-Réanimation, Assistance Publique, Hôpitaux de Paris, Paris, France. [19]These authors contributed equally: Raphael Cohen, Vincent Duval. ✉e-mail: hafid.aitoufella@inserm.fr

