## [Transparent Peer Review file · Nature Communications]

NK cells promote cardiac cell death and regulate myelopoiesis in myocardial infarction

Corresponding Author: Professor Hafid Ait-Oufella

Version 0:

Reviewer comments:

Reviewer #1

(Remarks to the Author)

Current data on the role of NK cells in myocardial ischemia are generally insufficient, and the prevailing hypothesis is that NK cells, early after myocardial ischemia, are detrimental due to their promotion of excessive inflammation and direct cytotoxic effects. Nevertheless, at later stages, NK cell activity appears to exert beneficial effects by facilitating the clearance of dead cells and promoting repair.

The study by Raphael Cohen et al. sheds new light by further corroborating the detrimental role of NK cells in the setting of acute ischemia in a mouse model of myocardial infarction induced by permanent coronary artery ligation. The authors also present an analysis of human myocardial tissue that correlates with their findings in mice.

Unfortunately, the study remains largely descriptive, despite employing several elegant *in vivo* approaches. It lacks a clear elucidation of the cellular mechanisms underlying the detrimental effects of NK cells in the acute phase of myocardial infarction. Most of the experiments rely on monoclonal antibodies and site-specific recombinases, which are often treated as foolproof tools but in reality require rigorous controls to ensure true specificity for the targeted cells.

The main weakness of this manuscript lies in the insufficient exploration of cellular and molecular mechanisms. The detrimental role of NK cells is correlated to Granzyme B activity; however, this merely replicates findings previously reported by the same group a few years ago. One would instead expect NK cells to modulate the broader immune response within the ischemic environment, necessitating analysis of other immune cells such as macrophages and monocytes, as suggested by the CCR2 experiments. Cell-to-cell immune interactions should therefore be systematically assessed. Additionally, the mechanisms through which NK cells influence infarct size remain unexplored. If the effect is mediated by cardiomyocyte apoptosis, this should be directly demonstrated. Similarly, the impact of NK modulation on interstitial fibrosis should be clarified—does it involve direct interactions between NK cells and fibroblasts?

Another important limitation concerns the specificity of the monoclonal antibodies used to deplete NK cells. Specifically, NK1.1 is not exclusively expressed on NK cells. Invariant NKT (iNKT) cells—a special subset of T cells expressing TCRs and recognizing lipid antigens—also express NK1.1. Moreover, under inflammatory conditions, some activated T cells (especially CD8⁺ T cells) can transiently upregulate NK1.1, and a subset of ILC1s (type 1 innate lymphoid cells) also expresses NK1.1. The CD8 depletion experiments are confusing, as the CD8 monoclonal antibody removes all CD8⁺ cells without discriminating between specific subsets.

A further important issue is that while it is plausible that removing a pro-inflammatory and pro-fibrotic cellular response improves myocardial remodeling, it remains unclear whether this is merely a secondary effect or whether it is associated with a true switch toward a pro-reparative, pro-regenerative response. Does NK cell depletion promote cardiomyocyte renewal and myocardial regeneration overall?

Finally, the human tissue analysis is purely descriptive. While it is understandable that human tissues have intrinsic limitations that make mechanistic studies difficult, there are human-based models, such as organoids, that could offer more suitable platforms to study the underlying molecular and cellular mechanisms in human myocardial tissues.

Reviewer #2

(Remarks to the Author)

Acute myocardial infarction (AMI) often leads to rapid patient mortality; however, our understanding of its immunopathological mechanisms is not as developed as in the field of tumor immunology. The manuscript's primary contribution lies in expanding the research scope of NK cells in pathological diseases. The authors aim to demonstrate that

NK cells play a destructive role in myocardial infarction (MI) tissue, directly leading to myocardial cell apoptosis and the generation of inflammatory myeloid cells. Despite this, the manuscript lacks rigor, which significantly limits its quality.

Major Comments:

Mechanism of NK cells vs CD8+ T cells in MI tissue: What is the difference in the mechanisms of NK cells and CD8+ T cells in MI tissue? Both are suggested to utilize GZMB as their key effector molecule. How do the authors exclude the roles of IFN- γ and TNF- α , especially in the induction of fibrosis?

NK cell migration to infarct tissue: How do NK cells in circulation sense the infarcted tissue? CCR2 appears to be a necessary but insufficient condition. More adhesion molecules should be considered. Additionally, can NK cells leave the infarct tissue and home back to the bone marrow to release GM-CSF? Based on the current results, GM-CSF levels in serum appear minimal.

Neutrophils in MI tissue: 70% of circulating human leukocytes are neutrophils. Why are they not considered as directly involved in the response to MI tissue, instead of relying on NK cells to release GM-CSF?

Targeting NK cells in MI therapy: The authors suggest targeting NK cells in MI treatment. According to results from the mouse MI model, NK cell counts peak only in the early stages (day 7, reaching ~75 cells/mg), and significantly decrease to baseline levels by day 14. However, heart function assessments are performed at days 21-28. Early MI is often difficult to detect.

Overall manuscript quality: The manuscript appears to be pieced together, and its rigor needs significant improvement.

Suggestions for Improving Rigor:

Figure 1C: Is "inf" an abbreviation for infarct? How was the infarct area identified in the fluorescence immunohistochemistry slices?

Figure 1D: The " $p < .001$ " is not standardized.

Figure 1G-H: Are the sequencing results from the subgroups in Figure 1I? If so, this introduces a logical order confusion. Additionally, for the non-myeloid single-cell transcriptomics, what molecular markers were used for sampling?

Figure 1P: There is no figure legend. The staggered offset could be misleading, and a histogram would be more suitable. More chemokines and adhesion molecules should be considered.

Figure 2B: Is the positioning of "INF" incorrect?

Figure 3K: The IgG control group is missing in the experimental design.

Figures 2G, 3J: No figure legends.

Figure 4A: Is this image reused?

Figures 4B-D: The authors perform transcript-level detection. How should we interpret the differences between sequencing and PCR sequencing? How should "Relative to Gapdh" be understood? Does it represent the percentage of GAPDH expression?

Figure 4GJK: What is the biological significance of the y-axis? How is myocardial damage assessed? Are the white areas necrotic? Why are they mainly concentrated in the vessel walls rather than the infarct area?

NK cell manipulation terminology: There is confusion in the terminology used for NK cell manipulation (e.g., NK cell depletion, ANTI-NK1.1, NK-KO, No-NK). The authors need to standardize and consistently use the correct terms.

Figure 4M: Data is missing.

Figures 4N-R: HL1 cells cannot be considered primary cardiomyocytes. The labeling of "Cardiomyocyte apoptosis" seems to be an over-interpretation. Additionally, how were NK cells pre-activated? Do pre-activated NK cells express GZMA? How was CASP3/7 activation detected? Is it phosphorylated CASP? Why are the time points so densely arranged?

Figure 5C: Is the y-axis correct?

Figure 5D: Was the cell loading amount consistent for flow cytometry? "Cellss" is a clear grammatical error.

Figure 5I: Is the mouse model used for MI?

Figure 5M: How were NK cells pre-stimulated?

Figure 6A: Were human NK cells isolated from the infarct tissue or peripheral blood? How were control samples obtained?

Figure 6D: What markers were used for NK cell staining?

Figure 6E: What is the biological significance of the "Mean expression" in the signaling pathway? How is it calculated?

Reviewer #3

(Remarks to the Author)

Comments to Authors:

The manuscript by Cohen et al., entitled "NK cells promote cardiac cell death and regulate myelopoiesis in myocardial infarction", investigates the role of NK cells in promoting apoptosis and impaired myocardial function. The authors demonstrate that treatment with an anti-NKG2A monoclonal antibody exacerbates ischemic heart failure. While the study addresses an important question, the manuscript can be strengthened by addressing the following concerns:

1. Figure 1: The authors demonstrate that increased infiltration of NK cells, notably CCR2+ NK cells, in the infarcted myocardium compared to sham. However, it remains unclear which specific chemokines mediate the recruitment of NK cells to the MI region, especially considering that monocyte-derived macrophages also express high levels of CCR2. Are the same chemokines responsible for recruiting CCR2-positive monocyte-derived macrophages to the infarct zone? Clarification is needed.

2. Figures 1/2: The specificity of NK cell depletion must be addressed. Since CCR2 and CCR5 are expressed by other immune cells, including macrophages. The use of a more specific NK cell marker, such as NKp46, would strengthen the study. This issue is evident in Figures 2b and 3d.

3. Figure 2: It would be informative to evaluate how NK cell depletion influences other immune cell populations and chemokine expression in MI.

4. As the authors are claiming that worse cardiac function and fibrosis are due to NK cell-mediated cytotoxicity, it'd be useful to demonstrate immune cell infiltration in the myocardium as well as increased cell death by TUNEL staining around day 3-7d following LAD ligation when the NK cells/mg heart tissue are at peak.

5. Can the authors discuss whether the interaction between NK cells and cardiomyocytes is direct or mediated via other cell types?

6. Figure 4: the immunostaining for GZMB shows significant GZMB even in NK depleted tissues. An explanation would be required. Additionally, details regarding co-culture conditions would be needed. Were the cells separated by inset/outlet or did they have any direct contacts? Since HL1 cells are proliferating

cardiomyocytes, they may not be an ideal model for cardiomyocyte cell death, which should be at least acknowledged.

7. Could the authors provide insight into whether human NK cells are likely to exhibit similar behavior in the context of MI? Regarding Figure 6, how did the authors differentiate active vs mature vs terminal NK? Also, I am having a very difficult time understanding what Figure 6e is trying to convey.

Minor comments

· Figure labeling: Ensure consistent formatting (e.g., Figure 1G – capital vs. lowercase letters).

· Page 6, Line 128: Revise "We then we investigated" to "We then investigated."

Reviewer #4

(Remarks to the Author)

Reviewer #5

(Remarks to the Author)

Version 1:

Reviewer comments:

Reviewer #1

(Remarks to the Author)

The authors have overall strengthened their mechanistic case that NK cells exacerbate acute post-MI remodeling via Granzyme B–dependent cardiomyocyte death and NK-derived GM-CSF–driven myelopoiesis. In the revised version of the manuscript, they provide additional controls addressing the specificity of NK depletion. They have also included new human data using 3D cardiac rings with human NK cells.

However, the human data, while improved, remain limited and largely correlative. The readout of the 3D rings is acute contractility over 12 hours only and does not relate directly to cardiac remodeling or cardiomyocyte death. In addition, there is no demonstration of Granzyme B involvement or of the specific NK-dependent pathways implicated in the mouse model. NK cells are derived from healthy donors, are IL-2–activated in vitro, and are not obtained from MI patients. These limitations are substantial and should be experimentally addressed.

Finally, while the authors argue that the role of NK cells in long-term cardiac remodelling remains to be fully elucidated and that addressing cardiac regeneration is outside the scope of this article, these points are not reflected in the revised manuscript. At a minimum, they should temper their “NK = bad” conclusion and explicitly acknowledge the possibility of later protective NK functions as a limitation and/or future direction. In parallel, they should discuss the possibility of cardiomyocyte renewal / cardiac regeneration as a potential contributing factor to the beneficial effects observed with acute NK depletion.

Reviewer #2

(Remarks to the Author)

The authors have made commendable improvements to the manuscript, and the overall quality has notably increased. However, we suggest the following revisions to further strengthen the work:

The discussion would benefit from a more detailed explanation of what initiates NK cell activation in the context of myocardial infarction. Please summarize available evidence indicating that NK cell activation occurs earlier than that of myeloid-derived pro-inflammatory cells.

Figure 4a, as it stands, appears repetitive and does not sufficiently stand on its own. We recommend either removing it or merging it into Figure 4b, with a corresponding note in the figure legend.

Reviewer #3

(Remarks to the Author)

The revised manuscript by Cohen et al., entitled “NK cells promote cardiac cell death and regulate myelopoiesis in myocardial infarction”, investigates the role of NK cells in promoting apoptosis and impaired myocardial function. Authors have adequately addressed our concerns.

Reviewer #4

(Remarks to the Author)

Reviewer #5

(Remarks to the Author)

Responses to reviewers

Reviewer #1 (Remarks to the Author):

Current data on the role of NK cells in myocardial ischemia are generally insufficient, and the prevailing hypothesis is that NK cells, early after myocardial ischemia, are detrimental due to their promotion of excessive inflammation and direct cytotoxic effects. Nevertheless, at later stages, NK cell activity appears to exert beneficial effects by facilitating the clearance of dead cells and promoting repair.

Response: We fully concur with the Reviewer that the role of NK cells has been insufficiently investigated. To the best of our knowledge, “the hypothesis that NK cells, early after myocardial ischemia, are detrimental“ has only been mentioned sporadically in general reviews addressing immune responses in MI, and, so far, has not been directly supported by experimental evidence, as demonstrated in our study.

We also appreciate the Reviewer’s reference to the potential beneficial effects of NK cells at later stages, which we presume relates to the study by Ayach et al. (Proc Natl Acad Sci U S A. 2006 Feb 14;103(7):2304-9). We would like to respectfully point out that this study used a rabbit anti-mouse polyclonal NK blocker from Cedarlane Laboratory, which is an anti-asialo GM1 antibody that is known to lack specificity for NK cells. Our laboratory has previously shown that anti-asialo GM1 depletes not only NK cells but also NKT cells and CD8+ T cells (see Table 1 in Nour-Eldine et al. Circ Res. 2018 Jan 5;122(1):47-57). Therefore, the conclusions from that study should, in our view, be interpreted with caution, given the non-specific nature of the depletion strategy employed.

Unfortunately, the study remains largely descriptive, despite employing several elegant in vivo approaches. It lacks a clear elucidation of the cellular mechanisms underlying the detrimental effects of NK cells in the acute phase of myocardial infarction. Most of the experiments rely on monoclonal antibodies and site-specific recombinases, which are often treated as foolproof tools but in reality require rigorous controls to ensure true specificity for the targeted cells.

Response: We would like to respectfully disagree with the reviewer’s comment. In our study, we employed multiple, complementary loss- and gain-of-function approaches to investigate, in a mechanistic manner, the critical role of NK cells in the context of MI. Specifically, we used genetically modified mouse models as well as pharmacological modulators in immunocompetent animals. Acknowledging the inherent limitations associated with each individual model, we intentionally combined several complementary experimental strategies to enhance the robustness and reliability of our findings.

The main weakness of this manuscript lies in the insufficient exploration of cellular and molecular mechanisms. The detrimental role of NK cells is correlated to Granzyme B activity; however, this merely replicates findings previously reported by the same group a few years ago.

Response: We agree with the Reviewer that our group has previously demonstrated the deleterious effect of CD8+-derived Granzyme B on post-MI cardiac remodeling. In the present study, we further show that granzyme B derived from NK cells also plays a major role in this process. However, the Reviewer suggests that we have insufficiently explored the cellular and molecular mechanisms involved. We respectfully disagree with this assessment, as our findings provide substantial evidence that:

- 1. NK cells contribute to deleterious post-ischemic cardiac remodeling independently of CD8+ T cells;**
- 2. NK cells promote cardiomyocyte cell death through the release of granzyme B;**

3. NK cells regulate myeloid cell trafficking by releasing GM-CSF within the bone marrow;
4. Mature cytotoxic NK cells are recruited into the myocardium post-MI, in part via CCR2; and
5. NK cells promote a pro-inflammatory phenotype in macrophages.

Importantly, Figures 3K-O specifically illustrate the pathogenic role of NK cells in the absence of CD8⁺ T cell activity, thereby supporting the conclusion that NK cell function in this context is independent of CD8⁺ T cells. Taken together, we believe that our study provides a novel mechanistic insight into how NK cells contribute to post-MI cardiac remodeling, a process that, to our knowledge, has not been previously characterized.

One would instead expect NK cells to modulate the broader immune response within the ischemic environment, necessitating analysis of other immune cells such as macrophages and monocytes, as suggested by the CCR2 experiments. Cell-to-cell immune interactions should therefore be systematically assessed.

Response : In Figure 4, we examined the impact of NK cell depletion or activation on local inflammatory responses following MI. In NK cell-depleted mice, mRNA levels of Il-1b and Tnfa were significantly reduced in infarcted hearts compared to controls (Figure 5a), along with a marked decrease in metalloproteinase activity (Figure 5b-c). NK cell depletion also led to reduced monocyte recruitment in the heart at days 5 and 7 post-MI (Figure 5d; Extended Data Fig.17 and 18a-b) and a lower proportion of CCR2⁺ macrophages at day 7 (Extended Data Fig.18c).

Conversely, NK cell activation using an anti-NKG2A mAb promoted a shift towards a more pro-inflammatory cardiac environment. This was characterized by elevated Il-1b and Il-6 mRNA levels, decreased Il-10 mRNA levels (Figure 5e), and increased local metalloproteinase activity (Figure 5f-g). Anti-NKG2A-mediated NK activation also enhanced the recruitment of neutrophils and monocytes in the heart at day 5 post-MI (Figure 5h; Extended Data Fig.18f-g) and increased the number of CCR2⁺ macrophages at day 7 (Extended Data Fig.18h).

Taken together, these data provide a comprehensive analysis of the local inflammatory response modulated by NK cell activity. In response to the reviewer's concern regarding cell-cell interactions, we performed scRNA-seq analysis on CD45⁺ cells isolated from ischemic heart tissue to further elucidate the cellular dynamics and interactions involved. As shown in the new supplemental figure 19, pseudobulk analysis revealed differentially expressed genes in monocyte and macrophage populations following NK cell depletion, and their expression was assessed across total cells per group. Notably, Cd36 and Hdac9 were upregulated, whereas P2rx7, which is involved in inflammasome activation, was downregulated in the NK-depleted group (Extended Data Fig.19e). Moreover, Tnf expression was reduced in the main populations of monocytes, macrophages and neutrophils in the depleted group (Extended Data Fig.19f). Altogether, these results indicate that NK cell depletion promotes a shift in macrophage profile toward a less inflammatory phenotype, highlighting the significant role of NK cells in modulating post-MI local immune responses.

Additionally, the mechanisms through which NK cells influence infarct size remain unexplored. If the effect is mediated by cardiomyocyte apoptosis, this should be directly demonstrated.

Response: We have substantial experience in quantifying apoptosis in cardiac tissue in the context of MI. In our view, TTC staining remains the most comprehensive and integrative method for evaluating tissue necrosis in the heart. This approach is widely accepted and commonly used in the field to assess infarct size and myocardial injury (e.g.

Yang et al. *Circulation* 2006; Leuschner et al. *Circ Res* 2010; Santos-Zas et al. *Nat Comm* 2021).

To directly address the reviewer's concern, we conducted additional TUNEL staining experiments in two relevant models: NK cell depletion (using NK1.1 mAb) and NK cell activation (Anti-NKG2A). As shown in the new supplemental figure 12, we found that NK cell depletion reduced TUNEL+ cells in the ischemic heart when compared to isotype treated mice, whereas anti-NKG2A mediated NK cell activation increased TUNEL + cells in the ischemic heart. These findings are fully consistent with the results from TTC staining further supporting the role of NK cells in modulating cardiomyocyte death following MI.

Similarly, the impact of NK modulation on interstitial fibrosis should be clarified—does it involve direct interactions between NK cells and fibroblasts ?

Response: In our models, interstitial fibrosis appears to follow the extent of infarct size and associated tissue necrosis. Specifically, we observed reduced fibrosis in conditions of NK cell depletion, where infarct size was smaller, and increased fibrosis following NK cell activation, which led to larger infarcts. Moreover, we found that modulation of the infarct size by NK cells also influenced the inflammatory signature in heart tissue, particularly the expression of key cytokines such as IL-1 β and TNF- α , which are well-established drivers of fibrotic remodeling. Notably, a recent study published in *Nature* (PMID: 39443792) demonstrated that IL-1 β plays a direct role in regulating fibroblast activity in the heart.

We agree that potential direct interactions between NK cells and fibroblasts in the setting of MI are of interest and are of great interest. While exploring this question is beyond the scope of the current study, we hope the Reviewer will appreciate our focus on other mechanistic aspects. Importantly, we have highlighted this point in the discussion section of the revised manuscript as a promising avenue for future research.

Another important limitation concerns the specificity of the monoclonal antibodies used to deplete NK cells. Specifically, NK1.1 is not exclusively expressed on NK cells. Invariant NKT (iNKT) cells—a special subset of T cells expressing TCRs and recognizing lipid antigens—also express NK1.1. Moreover, under inflammatory conditions, some activated T cells (especially CD8+ T cells) can transiently upregulate NK1.1, and a subset of ILC1s (type 1 innate lymphoid cells) also expresses NK1.1.

Response: We appreciate the Reviewer's comment regarding NKT cells. While NKT cells do express NK1.1, this population is minimal in the context of murine MI, approximately 10-fold lower than NK cells, as shown in Extended Data Fig.2. Importantly, anti-NK1.1 mAb treatment did not affect the CD8+ T population, and these data are now included as a new Supplemental figure 7d in the revised manuscript.

Regarding ILC1s, our scRNA-seq analysis revealed that this population is extremely rare and virtually undetectable in heart tissue under ischemic conditions, suggesting that they are unlikely to contribute significantly to the observed phenotype.

As noted in the manuscript, we are fully aware that anti-NK1.1 is not entirely specific for NK cells. To address this limitation and minimize potential confounding factors, we employed multiple complementary approaches. Specifically, we used Nkp461^{Cre+/-}R26R^{DTA} mice, which allow for specific depletion of NK cells, and further confirmed NK cell-specific effects through repopulation experiments with purified NK cells. These complementary strategies reinforce the specificity and validity of our conclusions regarding NK cell function in post-ischemic cardiac remodeling.

The CD8 depletion experiments are confusing, as the CD8 monoclonal antibody removes all CD8+ cells without discriminating between specific subsets.

Response: In the context of MI, our scRNA-seq analysis at day 5 post-MI showed that CD8 α expression was restricted to CD3+ CD8+ T cells and a very small subset of dendritic cells. Importantly, we did not detect significant CD8 α expression in other cardiac immune cell populations at this stage. Based on these observations, we consider our anti-CD8 depleting strategy to be both valid and appropriately specific for targeting CD8+ T cells in this experimental context.

A further important issue is that while it is plausible that removing a pro-inflammatory and pro-fibrotic cellular response improves myocardial remodeling, it remains unclear whether this is merely a secondary effect or whether it is associated with a true switch toward a pro-reparative, pro-regenerative response. Does NK cell depletion promote cardiomyocyte renewal and myocardial regeneration overall ?

Response: We appreciate the Reviewer's comment regarding the potential role of NK cells in cardiac regeneration. While this is indeed an intriguing area of research, it falls outside the scope of the current study, which was focused specifically on the contribution of NK cells to acute injury and adverse post-schemic remodeling. Furthermore, the hypothesis that NK cells may contribute to heart repair following MI is primarily based on the study by Ayach et al. (PNAS 2006), which should be interpreted with caution due to the non-specific depletion strategy employed, as discussed above.

Finally, the human tissue analysis is purely descriptive. While it is understandable that human tissues have intrinsic limitations that make mechanistic studies difficult, there are human-based models, such as organoids, that could offer more suitable platforms to study the underlying molecular and cellular mechanisms in human myocardial tissues.

Response: We thank the Reviewer for raising the important point regarding the human relevance of our findings. We were fortunate to access rare human heart tissue samples from MI patients, which allowed us to identify and characterize the local NK cell population. Using spatial transcriptomic analysis, we were able to describe both the phenotype and spatial distribution of NK cell subsets over time across distinct cardiac regions.

While cardiac organoids may become a valuable tool in the future, they are not fully established for studying human cardiac pathophysiology, particularly with respect to immune cell-tissue interactions. Current organoid systems do not recapitulate the complex in vivo interactions between hematopoietic cells and cardiac tissue, which are essential for investigating the role of NK cells during myocardial infarction.

To further address this point, we co-cultured human 3D cardiac ring organoids with purified human NK cells and measured contractility parameters before and 12 hours after co-culture (new Figure 4r & Extended Data Fig.15a). Non-activated NK cells had no effect on cardiac organoid function, whereas activated NK cells impaired both contractility and relaxation in a dose-dependent manner (new Figure 4s & Extended Data Fig.15b), providing experimental support for the relevance of our findings in the human setting.

Reviewer #2 (Remarks to the Author):

Acute myocardial infarction (AMI) often leads to rapid patient mortality; however, our understanding of its immunopathological mechanisms is not as developed as in the field of tumor immunology. The manuscript's primary contribution lies in expanding the research scope of NK cells in pathological diseases. The authors aim to demonstrate that NK cells play a

destructive role in myocardial infarction (MI) tissue, directly leading to myocardial cell apoptosis and the generation of inflammatory myeloid cells. Despite this, the manuscript lacks rigor, which significantly limits its quality.

Major Comments:

Mechanism of NK cells vs CD8+ T cells in MI tissue: What is the difference in the mechanisms of NK cells and CD8+ T cells in MI tissue? Both are suggested to utilize GZMB as their key effector molecule.

Response: As shown in Supplemental figure 2, the kinetics of CD8+T cell and NK cell recruitment differ in the context of acute MI. CD8+ T/NK ratio was >1 at day 1 but became <1 at days 3-5-7. Both cell types exhibit cytotoxic activity via Granzyme B production. In addition, in this study we found that NK cells modulate myelopoiesis through the production of GM-CSF.

How do the authors exclude the roles of IFN- γ and TNF- α , especially in the induction of fibrosis ?

Response: We thank the reviewer for his/her insightful comment regarding cytokines and fibrosis, which have been further elaborated in the Discussion section.

In our work, fibrosis closely followed infarct size and the extent of myocardial necrosis. In conditions where infarct size was reduced (i.e. following NK depletion), fibrosis was also decreased. Conversely, in settings where infarct size was exacerbated (i.e. after NK activation), we observed an increase in fibrosis. In addition, we found that the modulation of infarct size correlated with the inflammatory profile in cardiac tissue, particularly IL-1 β expression. Specifically, Il1b mRNA levels were decreased in the NK depletion model and increased in the NK activation model. These findings are in line with a recent study published in Nature (PMID: 39443792), which demonstrated that IL-1 β regulates fibroblast activity in the heart. Interestingly, although fibrosis was elevated in the NK cell activation experiment, we did not observe any significant change in Tnfa mRNA expression (figure 5e) suggesting that TNF- α may not play a central role in fibrosis in our experimental models.

To address the reviewer's comment regarding the role of IFN- γ , we have quantified T-bet mRNA levels in both the NK depletion and NK activation experiments, Tbet being a key regulator of IFN- γ production (Huang et al Front Immunol 2021). As shown in the new extended data figure 16, in experiments with NK cell activation (anti-NKG2A), we did not observe any change in Tbet gene expression between isotype treated and anti-NKG2A-treated mice. Given that fibrosis was more pronounced in the heart of anti-NKG2A-treated mice, these results suggest that IFN- γ may not play a central role in fibrosis in our experimental models. These additional results have been added in the discussion section of the revised manuscript.

NK cell migration to infarct tissue: How do NK cells in circulation sense the infarcted tissue? CCR2 appears to be a necessary but insufficient condition. More adhesion molecules should be considered.

Response: As shown in Supplemental fig.4, we used scRNA-seq to analyze the expression of 22 chemokine receptors in infiltrating leukocytes. We found that, in the context of MI, infiltrating NK cells exclusively expressed CCR2, CCR5 and CXCR4. This observation led us to specifically investigate the roles of CCR2 and CCR5 in NK cell recruitment, using KO mouse models. We found that Ccr2 global deletion led to reduced NK cell infiltration in the ischemic heart at day 3. Given that global CCR2 deletion impacts on

several immune population, we performed an additional series of experiments to support the critical role of CCR2 for NK cell recruitment in the ischemic heart tissue. We repopulated Nkp46iCre^{+/-} R26RDTA (referred to as NK^{KO}) mice with purified Wild-type or Ccr2^{-/-} NK cells 3 days before MI (new Figure 1O). NK deficiency in NK^{KO} mice was validated by flow cytometry (Extended Data Fig.5j). As shown in figure 1P, at day 3 post-MI, NK cell repopulation in the blood was similar between groups. However, NK cell numbers were significant lower in the heart of MI-operated mice repopulated with Ccr2^{-/-} NK cells compared to those repopulated with Ccr2^{+/+} NK cells(New Figure 1QR). Finally, additional transcriptomic analyses performed on human tissue from MI patients confirmed that infiltrating NK cells express CCR2 (new Extended Data Fig.24)

Regarding the reviewer's concern about adhesion molecules, we described in the original version of the manuscript that a subset of mature NK cells, named NK1, which are recruited during the first days post-MI express high levels of CD11b (Itgam). CD11b an important member of the β 2-integrin subunit, which affects leukocyte migration and tissue infiltration. Additional analyses revealed that this N1 subset also expressed higher levels of CD49d (New Extended Data Fig 4).

To further address the reviewer's concern, we quantified Ccl2, Ccl3, Cxcl1 and Cxcl2 mRNA levels in the ischemic heart of controls NK-depleted (Anti-NK1.1) and NK-activated (Anti-NKG2A) groups. As shown in the new supplemental figure 16, NK depletion or NK activation had no effect on Ccl2, Ccl3, Cxcl1 and Cxcl2 mRNA levels. These new results have been added in the revised version of the manuscript.

Additionally, can NK cells leave the infarct tissue and home back to the bone marrow to release GM-CSF? Based on the current results, GM-CSF levels in serum appear minimal.

Response: We would like to thank the reviewer for this insightful question. Several hypothesis could be proposed to explain how NK cells modulate myelopoiesis. One possibility is NK cell recirculation from the ischemic heart to the BM (as suggested by the reviewer). Alternatively, NK cells may be activated directly within the BM by circulating factors. Parabiosis experiments have demonstrated that factors released during MI can modulate hematopoiesis (Cremer et al. JACC 2020). Other studies have proposed that DAMPs and/or cardiomyocyte-derived vesicles may also be involved. While we acknowledge these potential mechanisms, we believe that a thorough investigation of this question would require a separate, dedicated study.

Detecting and quantifying GM-CSF protein remains technically challenging. In our study, we observed that NK cells produce GM-CSF in the context of MI, based on protein detection in the blood and mRNA expression in the BM. More importantly, we investigated the functional relevance of NK cell-derived GM-CSF using Nkp46^{Cre^{+/-}} Csf2^{lox/lox} mice. Our results indicate that the selective deletion of GM-CSF in NK cells alters myelopoiesis and affects post-ischemic cardiac remodeling (figure 5). This in vivo experiment provides direct causal evidence supporting the role of NK cell-derived GM-CSF in the regulation of myelopoiesis.

Neutrophils in MI tissue: 70% of circulating human leukocytes are neutrophils. Why are they not considered as directly involved in the response to MI tissue, instead of relying on NK cells to release GM-CSF?

Response: In our study, we did not rule out a potential role for neutrophils in post-MI remodeling. Rather, we identified that NK cells contribute, at least in part, to the modulation of myelopoiesis. Specifically, NK cell depletion led to a reduction in monocyte content within the ischemic heart, whereas NK cell activation increased both monocyte

and neutrophil infiltration into cardiac tissue. Further analysis revealed that NK cells in the bone marrow produce GM-CSF, a key driver of myelopoiesis in the context of acute MI, but not in sham-operated mice (a new Supplemental figure 20 has been added to illustrate this point). Importantly, specific deletion of GM-CSF in NK cells led to reduced myelopoiesis and attenuated adverse post-ischemic cardiac remodeling.

Taken together, our data demonstrate for the first time that NK cells promote adverse cardiac remodeling through two complementary mechanisms:

1. Directly, by modulating cardiomyocyte apoptosis; and
2. Indirectly, by influencing myelopoiesis.

Targeting NK cells in MI therapy: The authors suggest targeting NK cells in MI treatment. According to results from the mouse MI model, NK cell counts peak only in the early stages (day 7, reaching ~75 cells/mg), and significantly decrease to baseline levels by day 14. However, heart function assessments are performed at days 21-28. Early MI is often difficult to detect.

Response: We found the reviewer's comment somewhat difficult to interpret. In our study, we demonstrated that NK cells are actively recruited into cardiac tissue following MI, with a peak infiltration occurring between days 5 and 7. Once in the heart, NK cells exacerbate local tissue damage. This early increase in cardiomyocyte death contributes to more severe cardiac dysfunction at later stages of post-MI remodeling.

From a therapeutic perspective, these findings suggest that targeting NK cells during the acute phase of MI could be beneficial. Specifically, mAb-mediated NK cell depletion initiated around the time of PCI and maintained during the first 10 days post-MI might offer a clinically relevant window. In humans, the onset of MI is typically easy to identify due to the presence of chest pain, and recent studies report that the average time between symptom onset and PCI is approximately 2 hours (ex :PMID: 39211964). We have clarified this point further in the revised Discussion section.

Suggestions for Improving Rigor:

Figure 1C: Is "inf" an abbreviation for infarct? How was the infarct area identified in the fluorescence immunohistochemistry slices?

Response: Immunofluorescent staining was performed to identify NKp46+ NK cells. The infarct zone was delineated based on tissue architecture, including alterations in nucleus morphology and the presence of immune cell infiltration.

Figure 1D: The " $p < .001$ " is not standardized.

Response : The manuscript has been revised accordingly to incorporate this modification.

Figure 1G-H: Are the sequencing results from the subgroups in Figure 1I? If so, this introduces a logical order confusion. Additionally, for the non-myeloid single-cell transcriptomics, what molecular markers were used for sampling?

Response : The figure has been adjusted accordingly. The scRNA-seq data originates from a previously published paper, where the identification of immune cell lineages in the cardiac immune infiltrate is described in detail [PMID: 35950218]. Briefly, immune cell lineages were defined based on the expression of canonical marker genes and surface markers (measured by CITE-seq) of mouse immune cells. In these data, a population of cells corresponding to T cells, NK cells and also comprising other lymphoid subsets (e.g. ILC2) formed a distinct cluster expressing markers of T and NK cells at the protein (e.g.

CD3, NK1.1) and transcriptomic (e.g. *Tbrc1*, *Nkg7*) level, and devoid of myeloid cell (e.g. *Ly6G*, *F4/80*) or B cell (*CD19*) markers. This subset of cells was here extracted and reanalyzed with higher granularity to reveal subsets of T and NK cells.

To address the reviewer's concern, the reference has been added in the manuscript.

Figure 1P: There is no figure legend. The staggered offset could be misleading, and a histogram would be more suitable. More chemokines and adhesion molecules should be considered.

Response: The figure has been replaced by a histogram for greater clarity.

As shown in Supplemental figure 5, we analyzed the expression of 22 chemokine receptors in heart tissue at day 5 post-MI using scRNA-seq. We found that infiltrating NK cells expressed only *CCR2*, *CCR5* and *CXCR4* mRNA. To assess the functional relevance of these receptors, we performed experiments using *CCR2* and *CCR5* KO mice. Our data indicate that *CCR2* contributes, at least in part, to NK cell recruitment into the heart. Unfortunately, *CXCR4* KO mice cannot be used in this context, as *CXCR4* is a critical regulator of global hematopoiesis and its global deletion is embryonically lethal.

Given that global *CCR2* deletion impacts on several immune population and to address the reviewer's concern, we performed an additional series of experiments to support the critical role of *CCR2* for NK cell recruitment in the ischemic heart tissue. We repopulated *Nkp46iCre^{+/-} R26RDTA* (referred to as *NK^{KO}*) mice with purified Wild-type or *Ccr2^{-/-}* NK cells 3 days before MI (new Figure 1O). NK deficiency in *NK^{KO}* mice was validated by flow cytometry (Extended Data Fig.5j). As shown in figure 1P, at day 3 after MI, NK cell repopulation in the blood was similar between groups. However, NK cell number was significantly lower in the heart of MI-operated mice repopulated with *Ccr2^{-/-}* NK cells compared to animals repopulated with *Ccr2^{+/+}* NK cells (New Figure 1QR).

To further address the reviewer's concern, we quantified *Ccl2*, *Ccl3*, *Cxcl1* and *Cxcl2* mRNA levels in the ischemic heart of controls NK depleted (Anti-NK1.1) and NK activated (Anti-NKG2A) groups. As depicted in the new supplemental figure 16, NK depletion or NK activation had no effect on *Ccl2*, *Ccl3*, *Cxcl1* and *Cxcl2* mRNA levels. These new results have been added in the revised version of the manuscript.

Figure 2B: Is the positioning of "INF" incorrect?

Response: Thank you for this comment. The position of "INF" has been modified accordingly.

Figure 3K: The IgG control group is missing in the experimental design.

Response: With all due respect, we disagree with the reviewer's concern. The objective of this experiment was to specifically assess the impact of NK cell activation in the absence of CD8⁺ T cells. Both experimental groups treated with an anti-CD8 depleting antibody to ensure effective CD8⁺ T cell depletion, which was confirmed by flow cytometry. The NK cell activated group additionally received an anti-NKG2A antibody, while the control group received an isotype control. Our data show that NK cell activation exerts a pathogenic effect in the absence of CD8⁺T cells.

Including an IgG control group would not address the specific question under investigation and would therefore not provide additional relevant information in this context.

Figures 2G, 3J: No figure legends.

Response: We apologize for the oversight. The figure legends have been updated accordingly.

Figure 4A: Is this image reused ?

Response: Indeed, this is the same UMAP visualization of scRNA-seq profiles from non-myeloid immune cells in figure 1i. This visualization is intended to assist readers in analyzing the results of Figure 4B results, specifically regarding the gene expression of Perforin, Granzyme A and Granzyme B.

Figures 4B-D: The authors perform transcript-level detection. How should we interpret the differences between sequencing and PCR sequencing? How should "Relative to Gapdh" be understood? Does it represent the percentage of GAPDH expression?

Response: ScRNA-seq (Fig 4B) enables the detection and quantification of transcripts in different cell populations. In figure 4CD, the quantification of mRNA levels in ischemic heart tissue was performed using qPCR at various time points post-MI. mRNA levels were normalized to Gapdh expression, with Gapdh serving as an internal control for gene expression.

Figure 4GJK What is the biological significance of the y-axis? How is myocardial damage assessed? Are the white areas necrotic? Why are they mainly concentrated in the vessel walls rather than the infarct area?

Response: As detailed in the Methods section and figure legends, 2,3,5-Triphenyltetrazolium chloride (TTC) staining was used to evaluate the infarct size at early time points (3 days after MI). TTC staining is a well-established method for identifying metabolically active tissue and is widely used to assess tissue viability (see Yang Circulation 2006; Leuschner et al. Circ Res 2010; Santos-Zas et al Nat Comm 2021). Necrotic tissue appears white after staining.

We are uncertain about the reviewer's comment regarding the vessel wall, as figures 4G, J and K clearly show left ventricle sections taken 3 days after MI.

To further support the pro-apoptotic role of NK cells, we quantified TUNEL+ cells in the ischemic heart at day 3 in CTR, NK depleted (anti-NK1.1) and NK activated (anti-NKG2A) treated mice. We found that NK depletion led to decreased number of TUNEL+ cells in the heart, whereas NK activation increased TUNEL+ cell numbers (See new supplemental fig12).

NK cell manipulation terminology: There is confusion in the terminology used for NK cell manipulation (e.g., NK cell depletion, ANTI-NK1.1, NK-KO, No-NK). The authors need to standardize and consistently use the correct terms.

Response : We apologize for the oversight. The terminology has been standardized accordingly.

Figure 4M: Data is missing.

Response: This mistake occurred during the PDF conversion of the PPT figure. It has been corrected accordingly.

Figures 4N-R: HL1 cells cannot be considered primary cardiomyocytes. The labeling of "Cardiomyocyte apoptosis" seems to be an over-interpretation. Additionally, how were NK cells pre-activated? Do pre-activated NK cells express GZMA? How was CASP3/7 activation detected? Is it phosphorylated CASP? Why are the time points so densely arranged?

Response: The figure legends have been modified accordingly: cardiomyocytes (HL1) apoptosis.

NK cells were pre-activated overnight with cytokines (rIL-15 at 5 ng/ml), then washed and co-cultured with HL1 cardiomyocytes. This procedure is described in the Methods section.

In our study, we did not investigate the role of Granzyme A in NK cell cytotoxic activity *in vitro*, as we found that NK cell depletion did not affect Gzma mRNA levels in ischemic heart tissue (see figure 4c).

Activated and non-activated NK cells were co-cultured with HL1 cardiomyocytes in different ratios (1:1, 1:5 and 1:10, based on cardiomyocyte count). After 24h of co-culture, NK cells were removed and HL1 cardiomyocytes were labeled with 5 μ M Caspase-3/7 Green Apoptosis Reagent (Incucyte, Welwyn Garden, United Kingdom), which was added to the growth medium. After a 30 min incubation at 37°C in the dark, apoptosis was evaluated in real time, using a Nikon Eclipse Ti microscope. Time-lapse images were captured at 10-min intervals. The IncuCyte Caspase-3/7 Green Apoptosis Assay is a live-cell imaging system that enables real-time, quantitative assessment of apoptosis in cell cultures. This assay uses a fluorogenic substrate that becomes fluorescent upon cleavage by caspase-3/7, key executioners of apoptosis.

Figure 5C: Is the y-axis correct?

Response: Sorry, this was an error, and the y-axis has been corrected to « fluorescent intensity » accordingly.

Figure 5D: Was the cell loading amount consistent for flow cytometry? "Cellss" is a clear grammatical error.

Response: The grammatical error has been corrected accordingly. Flow cytometry analysis after heart tissue digestion is routinely performed in our laboratory and others. We are able to detect 5.000 to 50.000 CD45+ cells per flow cytometry sample, depending on the timepoint after MI induction. This yields cells roughly 500 to 10.000 CD45+ cells per mg of cardiac tissue. Those numbers are numbers consistent with the literature (PMID: 30586758, 32901075, 35950218)

Figure 5I: Is the mouse model used for MI?

Response : Indeed, analysis has been done at day 3 post-MI. This point has been clarified in the legends

Figure 5M: How were NK cells pre-stimulated?

Response: As previously described by Louis et al. (PMID 32097462) who investigated the role of GM-CSF producing NK cells in experimental arthritis, NK cells were pre-stimulated with rIL-12 (50 microg/ml), rIL-15 (10 ng/ml) and rIL-18 (50ng/ml). This procedure has been clarified in the Methods section.

Figure 6A: Were human NK cells isolated from the infarct tissue or peripheral blood? How were control samples obtained?

Response: As detailed in the Methods section, we measured the global transcriptomic signature of heart tissue sections using Nanostring technology. Subsequently, bioinformatic analysis was performed to identify NK cell-related pathways. Normal heart tissue was obtained from Creative Bioarray (United States).

Figure 6D: What markers were used for NK cell staining ?

Response: For immunostaining, we used an anti-NKp46 monoclonal mAb (clone 8E5B, Innate Pharma).

Figure 6E: What is the biological significance of the "Mean expression" in the signaling pathway? How is it calculated?

Response: The mean expression of pathway represents the “calculated module scores using the AddModuleScore function in Seurat”. For each cell, the average expression of a given gene set (here, genes associated with inflammatory and cytotoxic pathways), is computed relative to control genes of similar expression. This yields a relative activity score for the pathway in each cell. We used 50 randomly selected control genes to normalise the score.

Biological significance: The module score provides a measure of the transcriptional activity of the cytotoxic and inflammatory pathways in each cell. This allows us to detect subtle but significant shifts in pathway activity. Notably, the distribution of module scores across different time points and treatments highlights cell populations undergoing potential state transitions, thereby, showcasing the differential regulation of these pathways.

This explanation has been clarified in the revised manuscript.

Reviewer #3 (Remarks to the Author):

Comments to Authors:

The manuscript by Cohen et al., entitled “NK cells promote cardiac cell death and regulate myelopoiesis in myocardial infarction”, investigates the role of NK cells in promoting apoptosis and impaired myocardial function. The authors demonstrate that treatment with an anti-NKG2A monoclonal antibody exacerbates ischemic heart failure. While the study addresses an important question, the manuscript can be strengthened by addressing the following concerns:

1. Figure 1: The authors demonstrate that increased infiltration of NK cells, notably CCR2+ NK cells, in the infarcted myocardium compared to sham. However, it remains unclear which specific chemokines mediate the recruitment of NK cells to the MI region, especially considering that monocyte-derived macrophages also express high levels of CCR2. Are the same chemokines responsible for recruiting CCR2-positive monocyte-derived macrophages to the infarct zone? Clarification is needed.

Response: As shown in Supplementary Figure 5A and well documented in the literature, each cell type expresses a distinct set of chemokine receptors, although several receptors are shared among different cell types. For example, CCR2 is expressed by both NK cells and classical monocytes.

Given that global CCR2 deletion impacts on several immune population and to address the reviewer’s concern, we performed an additional series of experiments to support the critical role of CCR2 for NK cell recruitment in the ischemic heart tissue. We repopulated Nkp46iCre^{+/-} R26RDTA (referred to as NK^{KO}) mice with purified Wild-type or Ccr2^{-/-} NK cells 3 days before MI (new Figure 1O). NK deficiency in NK^{KO} mice was validated by flow cytometry (Extended Data Fig.5j). As shown in the new figure 1P, at day 3 post-MI, NK cell repopulation in the blood was similar between groups. However, NK cell number was significant lower in the heart of MI-operated mice repopulated with Ccr2^{-/-} NK cells compared to animals repopulated with Ccr2^{+/+} NK cells (New Figure 1QR). These results confirmed the key role of CCR2 in the recruitment of NK cells in the ischemic heart.

2. Figures 1/2: The specificity of NK cell depletion must be addressed. Since CCR2 and CCR5 are expressed by other immune cells, including macrophages. The use of a more specific NK cell marker, such as NKp46, would strengthen the study. This issue is evident in Figures 2b and 3d.

Response: As shown in figure 1a, NK cells were defined for flow cytometry analysis as CD11b⁻CD3⁻NK1.1⁺ NKp46⁺. For immunostainings, NK cells were identified as NKp46⁺ cells (yellow staining).

3. Figure 2: It would be informative to evaluate how NK cell depletion influences other immune cell populations and chemokine expression in MI.

Response: To address the reviewer's concern, we performed scRNA-seq analysis on CD45⁺ cells isolated from ischemic heart tissue to further elucidate the cellular dynamics and interactions involved. As shown in the new supplemental figure 19, pseudobulk analysis revealed differentially expressed genes in monocyte and macrophage populations following NK cell depletion, assessed across total cells per group. Of note, Cd36 and Hdac9 were upregulated, whereas P2rx7, involved in inflammasome activation, was downregulated in the NK depleted group (Extended Data Fig.19e). Moreover, Tnf expression in the main monocyte, macrophage and neutrophil populations was reduced upon NK cell depletion (Extended Data Fig.19f). Altogether, these results indicate that NK cell depletion promotes a shift in macrophage profiles toward a less inflammatory phenotype.

To further address the reviewer's concern, we quantified Ccl2, Ccl3, Cxcl1 and Cxcl2 mRNA levels in the ischemic heart of control, NK depleted (Anti-NK1.1) and NK activated (Anti-NKG2A) groups. As shown in the new supplemental figure 16, NK depletion or NK activation had no impact on Ccl2, Ccl3, Cxcl1 and Cxcl2 mRNA levels. These new results have been added in the new version of the manuscript.

4. As the authors are claiming that worse cardiac function and fibrosis are due to NK cell-mediated cytotoxicity, it'd be useful to demonstrate immune cell infiltration in the myocardium as well as increased cell death by TUNEL staining around day 3-7d following LAD ligation when the NK cells/mg heart tissue are at peak.

Response: We have extensive experience in quantifying apoptosis in heart tissue in the context of MI, and consider that TTC staining the most comprehensive and integrative method for assessing cardiac tissue necrosis and viability (e.g. Yang Circulation 2006; Leuschner et al. Circ Res 2010; Santos-Zas et al Nat Comm 2021). In previous studies, we performed TUNEL staining and observed very few TUNEL⁺ cells in the heart after MI, as the transition from apoptosis to necrosis occurs rapidly.

However, to address the reviewer's concern, we performed TUNEL staining at day 3 post-MI in both NK depletion (NK1.1) and NK activation (anti-NKG2A) groups. NK depletion was associated with a significant reduction of TUNEL⁺ cells in the ischemic heart, whereas NK activation was led to an increase, consistent with the TTC staining results (New supplemental figure 12).

5. Can the authors discuss whether the interaction between NK cells and cardiomyocytes is direct or mediated via other cell types ?

Response: As suggested by the reviewer, this question has been discussed in detail in the Discussion section. We have gathered substantial evidence demonstrating that NK cells directly promote cardiomyocyte death through Granzyme B release in the context of MI, supported by both in vitro and in vivo experiments. Moreover, since we have shown that NK cells also contribute to myelopoiesis during MI and that myeloid cells exhibit pro-

apoptotic activity, we propose that NK cells exert an indirect pro-apoptotic effect through the regulation of myelopoiesis.

6. Figure 4: the immunostaining for GZMB shows significant GZMB even in NK depleted tissues. An explanation would be required.

Response: As detailed in the manuscript, the detection of GZMB+ cells in the ischemic heart of NK-depleted mice is attributed to the presence of CD8+ T cells, which also produce GZMB (see Santos-Zas et al. Nat Comm 2021).

Additionally, details regarding co-culture conditions would be needed. Were the cells separated by inset/outlet or did they have any direct contacts? Since HL1 cells are proliferating cardiomyocytes, they may not be an ideal model for cardiomyocyte cell death, which should be at least acknowledged.

Response: To address the reviewer's concern, the description of the co-culture method has been revised. The cells were not separated by a membrane, allowing for direct cell-cell contact interactions. Furthermore, we demonstrated a direct pro-apoptotic role of Granzyme B, a soluble protein, as cardiomyocyte death was reduced in the co-cultures with GzmB^{-/-} NK cells.

7. Could the authors provide insight into whether human NK cells are likely to exhibit similar behavior in the context of MI? Regarding Figure 6, how did the authors differentiate active vs mature vs terminal NK? Also, I am having a very difficult time understanding what Figure 6e is trying to convey.

Response: In figure 6, NK cell subsets were classified based on previously published work (Yang et al Nat Com 2019 PMID: 31477722). Active NK cells were defined by a higher module score, characterized by up-regulated DEGs associated with "Inflamed NK" cells, including several immediate early genes (IEGs), such as NR4A2, DUSP1, FOSB, FOS, JUN, and JUNB. Mature NK cells were characterized by high expression of molecules important for cytotoxic function and cytolytic molecules, including perforin (PRF1) and granzymes (GZMA, GZMB, GZMH). Terminal NK cells express functional molecules similar to the "Mature NK" subset but show high expression of the transcription factor ZEB2, indicating terminal maturity. This clarification has been clarified in the revised manuscript.

Figure 6E is based on spatial transcriptomics of human heart tissue at different time points after MI. Using this method and bioinformatic analysis, we were able to characterize NK subsets in different areas over time. We found that NK cells with an inflammatory and cytotoxic profile were detected at early stages in both the ischemic and remote zones. At later stages, NK cells with a cytotoxic profile were detected in infarcted areas. To address the reviewer's concern, the results in figure 6 have been described in greater detail in the revised manuscript.

Minor comments

Figure labeling: Ensure consistent formatting (e.g., Figure 1G – capital vs. lowercase letters).

Response: The modification has been made accordingly.

Page 6, Line 128: Revise "We then we investigated" to "We then investigated."

Response: The modification has been made accordingly.

Responses to reviewers

Reviewer #1 (Remarks to the Author):

The authors have overall strengthened their mechanistic case that NK cells exacerbate acute post-MI remodeling via Granzyme B–dependent cardiomyocyte death and NK-derived GM-CSF–driven myelopoiesis. In the revised version of the manuscript, they provide additional controls addressing the specificity of NK depletion. They have also included new human data using 3D cardiac rings with human NK cells.

Response : We would like to thank the reviewer for his/her comments that helped to improve the manuscript.

However, the human data, while improved, remain limited and largely correlative. The readout of the 3D rings is acute contractility over 12 hours only and does not relate directly to cardiac remodeling or cardiomyocyte death. In addition, there is no demonstration of Granzyme B involvement or of the specific NK-dependent pathways implicated in the mouse model. NK cells are derived from healthy donors, are IL-2–activated in vitro, and are not obtained from MI patients. These limitations are substantial and should be experimentally addressed.

Response : Our in vitro experiments using 3D cardiac rings showed that activated NK rapidly impaired human cardiomyocyte functions, consistent with all the results obtained from murine models and co-culture experiments. Regarding the underlying mechanisms, we acknowledge the reviewer's concern about demonstrating granzyme B involvement in the human system. To the best of our knowledge, functional blocking or neutralizing antibodies against human granzyme B are not commercially available as research tools. While detection antibodies exist, validated functional inhibitors have not been developed commercially. Alternative approaches, such as small-molecule inhibitors would require substantial optimization and validation within our 3D tissue model. We also acknowledge that NK cells derived from healthy donors may not fully recapitulate the phenotype of NK cells from MI patients. However, obtaining sufficient numbers of viable NK cells from MI patients for functional assays presents significant logistical and ethical challenges. In France, ethical approval for such experiments would require approximately 8-9 months.

To address the reviewer's concern, an additional limitation has been added to the discussion section (Page 18).

Finally, while the authors argue that the role of NK cells in long-term cardiac remodelling remains to be fully elucidated and that addressing cardiac regeneration is outside the scope of this article, these points are not reflected in the revised manuscript. At a minimum, they should temper their “NK = bad” conclusion and explicitly acknowledge the possibility of later protective NK functions as a limitation and/or future direction. In parallel, they should discuss the possibility of cardiomyocyte renewal / cardiac regeneration as a potential contributing factor to the beneficial effects observed with acute NK depletion.

Response : To address the reviewer's concern, the discussion has been modified as follows: « *Our study highlights a global pathogenic role for NK cells in the immune-inflammatory responses contributing to tissue damage following acute MI. However, we cannot rule out a protective role for specific NK cell subsets at later stages, as previous studies have reported pro-angiogenic activities of NK cells {Bouchentouf, 2010 #38569}* » (Page 21).

Reviewer #2 (Remarks to the Author):

The authors have made commendable improvements to the manuscript, and the overall quality has notably increased.

Response : We would like to thank the reviewer for his/her comments that helped to improve the manuscript.

However, we suggest the following revisions to further strengthen the work. The discussion would benefit from a more detailed explanation of what initiates NK cell activation in the context of myocardial infarction. Please summarize available evidence indicating that NK cell activation occurs earlier than that of myeloid-derived pro-inflammatory cells.

Response : The discussion regarding factors that may activate NK cells in the context of myocardial infarction has been expanded in the Discussion section (Page 21). To our knowledge, no published data currently explore the very early timing of NK versus myeloid cell activation following cardiac tissue ischemia. Further studies will be required to specifically address this point.

Figure 4a, as it stands, appears repetitive and does not sufficiently stand on its own. We recommend either removing it or merging it into Figure 4b, with a corresponding note in the figure legend.

Response : Figure 4a has been modified accordingly

Reviewer #3 (Remarks to the Author):

The revised manuscript by Cohen et al., entitled “NK cells promote cardiac cell death and regulate myelopoiesis in myocardial infarction”, investigates the role of NK cells in promoting apoptosis and impaired myocardial function. Authors have adequately addressed our concerns.

Response : We would like to thank the reviewer for his/her comments that helped to improve the manuscript.

Reviewer #4 (Remarks to the Author):

Response : We would like to thank the reviewer for his/her comments that helped to improve the manuscript.

Reviewer #5 (Remarks to the Author):

Response : We would like to thank the reviewer for his/her comments that helped to improve the manuscript.